# Sustained expression of *unc-4* homeobox gene and *unc-37/Groucho* in postmitotic neurons specifies the spatial organization of the cholinergic synapses in *C. elegans*

Mizuki Kurashina[1,2], Jane Wang[1], Jeffrey Lin[1], Kathy Kyungeun Lee[1], Arpun Johal[1], Kota Mizumoto[1,2,3,4]*

[1]Department of Zoology, University of British Columbia, Vancouver, Canada; [2]Graduate Program in Cell and Developmental Biology, University of British Columbia, Vancouver, Canada; [3]Life Sciences Institute, University of British Columbia, Vancouver, Canada; [4]Djavad Mowafaghian Centre for Brain Health, University of British Columbia, Vancouver, Canada

**Abstract** Neuronal cell fate determinants establish the identities of neurons by controlling gene expression to regulate neuronal morphology and synaptic connectivity. However, it is not understood if neuronal cell fate determinants have postmitotic functions in synapse pattern formation. Here we identify a novel role for UNC-4 homeobox protein and its corepressor UNC-37/Groucho, in tiled synaptic patterning of the cholinergic motor neurons in *Caenorhabditis elegans*. We show that *unc-4* is not required during neurogenesis but is required in the postmitotic neurons for proper synapse patterning. In contrast, *unc-37* is required in both developing and postmitotic neurons. The synaptic tiling defects of *unc-4* mutants are suppressed by *bar-1/β-catenin* mutation, which positively regulates the expression of *ceh-12/HB9*. Ectopic *ceh-12* expression partly underlies the synaptic tiling defects of *unc-4* and *unc-37* mutants. Our results reveal a novel postmitotic role of neuronal cell fate determinants in synapse pattern formation through inhibiting the canonical Wnt signaling pathway.

*For correspondence:
mizumoto@zoology.ubc.ca

**Competing interests:** The authors declare that no competing interests exist.

## Introduction

Formation of a functional neuronal circuit starts from the establishment of individual neuronal identities. The establishment and maintenance of neuronal cell fates are governed by a series of transcription factors called neuronal cell fate determinants. These neuronal cell fate determinants regulate the gene expression profile that defines the functional traits and properties of the neuron, such as neurotransmitter choice and neuronal morphology (*Allan and Thor, 2015*; *Hobert, 2016*; *Hobert and Kratsios, 2019*). For example, a conserved LIM (LIN-11, Isl-1, MEC-3) homeobox transcription factor Isl1 determines cholinergic neuronal cell fates in the mice and chick spinal cord and hindbrain (*Cho et al., 2014*; *Ericson et al., 1992*; *Liang et al., 2011*; *Pfaff et al., 1996*; *Tsuchida et al., 1994*). Similarly, the *Drosophila* homolog of the vertebrate Isl1, *islet*, specifies motor neurons and interneurons in the ventral nerve cord (*Thor and Thomas, 1997*). Furthermore, the HB9/MNR2 transcription factor is required for the differentiation and specification of motor neurons in chick and mice (*Arber et al., 1999*; *Tanabe et al., 1998*). In *Caenorhabditis elegans*, MEC-3 and UNC-86, which are LIM and POU (Pit-1/Oct1/2/UNC-86) homeobox proteins, respectively, function cooperatively to specify the glutamatergic mechanosensory neurons (*Duggan et al., 1998*; *Finney et al., 1988*; *Way and Chalfie, 1988*; *Xue et al., 1993*).

Interestingly, many neuronal cell fate determinants are continuously expressed in postmitotic neurons (*Deneris and Hobert, 2014*). Consistent with this sustained expression, several studies have shown that neuronal cell fate determinants are crucial for the neuronal maturation and maintenance of the respective cell fates. For example, the ETS (Erythroblast Transformation Specific) domain transcription factor PET-1, which determines the serotonergic neuronal fates in the Raphe nuclei of the brainstem of mice, is continuously expressed (*Hendricks et al., 1999*) and controls the neuronal maturation of serotonergic neurons via controlling the expression of the neuronal maturation factor, Engrailed 1 (*Wyler et al., 2016*). In *C. elegans*, the COE (Collier/Olf/EBF) type transcription factor, UNC-3, is required for both specification and maintenance of the cholinergic motor neuron identities (*Kerk et al., 2017*; *Kratsios et al., 2012*). The POU homeobox protein UNC-86 and its mammalian ortholog, BRN3A, are required for the maintenance of the cell fates of several sensory neurons in *C. elegans* and the medial habenular neurons in the diencephalon of mice (*Serrano-Saiz et al., 2018*).

Several studies have also revealed the role of neuronal cell fate determinants in neuronal wiring and synaptic specificity. For example, *Drosophila* HB9 is required in a subset of ventrally projecting motor neurons and the serotonergic EW1, EW2, and EW3 interneurons for axon pathfinding and target specificity (*Odden et al., 2002*). In *C. elegans*, in the mutants of *unc-55*, which encodes an ortholog of COUP-TF (chicken ovalbumin upstream promoter transcription factor), the ventrally innervating D-class (VD) GABAergic motor neurons innervate the dorsal body wall muscles instead of the ventral body wall muscles, due to the ectopic expression of the genes which are normally expressed only in the dorsally innervating D-class (DD) neurons (*Howell et al., 2015*; *Shan et al., 2005*; *Zhou and Walthall, 1998*). In *unc-3* mutants, the AVA interneurons lose the majority of the synaptic connections with the genuine targets and, in turn, form ectopic synaptic connections with unidentified neurons (*Pereira et al., 2015*). While functions of neuronal cell fate determinants in neuronal wiring and synaptic specificity have been identified, little is known about the postmitotic roles of neuronal cell fate determinants in neuronal wiring.

The paired-type homeobox transcription factor, UNC-4, specifies the ventral and dorsal A-type cholinergic motor neurons (VAs and DAs) by repressing the alternative B-type neuronal fates (VBs and DBs) in *C. elegans* (*Kerk et al., 2017*; *Kratsios et al., 2012*; *Winnier et al., 1999*). The repression of the B-type fates requires the physical interaction between UNC-4 and the Groucho-like transcriptional corepressor, UNC-37, to repress the expression of B-type-specific genes such as *acr-16/nicotinic acetylcholine receptor* and *acr-5/acetylcholine receptor α-like subunit* (*Kerk et al., 2017*; *Winnier et al., 1999*). While the loss of *unc-4* and *unc-37* and the subsequent ectopic expression of B-type genes does not alter the morphology of A-type motor neurons, this results in the altered connectivity of the A-type motor neurons from the upstream command interneurons (*Miller et al., 1992*; *White et al., 1992*). In wild-type animals, the VA motor neurons receive presynaptic input from the AVA command interneurons (*White et al., 1986*). However, in the loss of function mutants of *unc-4* and *unc-37*, the VA neurons do not form synaptic connections with the AVA interneurons but instead form synaptic connections with the AVB command interneurons, which normally innervate the VB motor neurons (*Miller et al., 1992*; *White et al., 1992*; *Winnier et al., 1999*). Temporal knockdown experiments using a temperature-sensitive mutant of *unc-4* showed that it is required for proper AVA>VA connectivity, primarily in the late first larval (L1) stage when the VA neurons are born, suggesting that this miswiring is likely due to the partial cell fate transformation of the A-type motor neurons to the B-type motor neurons (*Miller et al., 1992*). However, the knockdown of *unc-4* at L2-L3 also caused partial AVA>VA wiring defect, suggesting that *unc-4* is also required after the AVA>VA wiring completes. This observation implies that *unc-4* also functions in the maintenance of AVA>VA synaptic connection (*Miller et al., 1992*). Recent work in *Drosophila* has identified the role of Unc-4 to promote cholinergic fate and to inhibit GABAergic fate during development (*Lacin et al., 2020*), suggesting a conserved role of *unc-4* in neuronal cell fate specification.

In this study, we report a novel postmitotic role for *unc-4* and *unc-37* in precise synapse pattern formation in the DA class cholinergic motor neurons. DA neurons exhibit a unique 'tiled' synaptic innervation pattern, in which the synaptic domain of a DA neuron does not overlap with those from the neighboring DA neurons (*White et al., 1976*; *Figure 1A*). Previously, we have shown that the interaxonal interaction between two DA neurons (DA8 and DA9) mediated by Semaphorin–Plexin signaling establishes the synaptic tiling of DA8 and DA9 (*Chen et al., 2018*; *Mizumoto and Shen, 2013a*). In the loss of function mutants of *unc-4* and *unc-37*, we show that the tiled synaptic innervation between DA8 and DA9 is severely disrupted, similar to the mutants of Semaphorin-Plexin

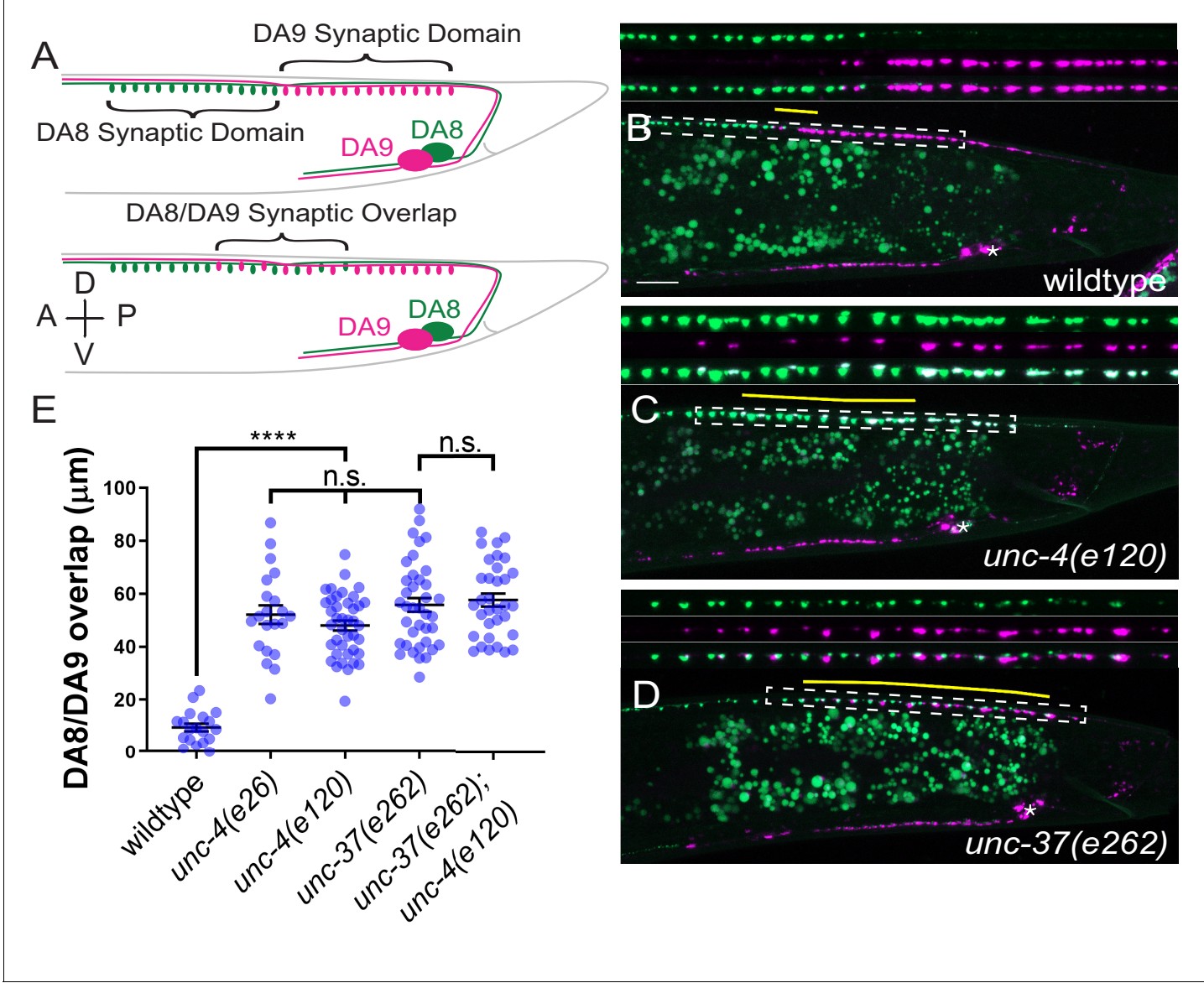

**Figure 1.** *unc-4* and *unc-37* are required for tiled synaptic innervation of DA8 and DA9 neurons. (**A**) Schematic of the tiled innervation between DA8 and DA9 neurons of wild-type (top) and synaptic tiling mutants (bottom). (**B–D**) Representative images of synaptic tiling in wildtype (**B**), *unc-4(e120)* (**C**), and *unc-37(e262)* (**D**) mutants. Magnified images (represented by the dotted box) of GFP, mCherry, and merged channels shown above. Synaptic overlap between the DA8 and DA9 synaptic domains are highlighted with yellow lines. Asterisks: DA9 cell body. Scale bar: 10 µm. (**E**) Quantification of overlap between DA8 and DA9 synaptic domains. See *Figure 1—source data 1*. Each dot represents a single animal. Black bars indicate mean ± SEM. n.s.: not significant; *: p<0.05; ****: p<0.0001.

The online version of this article includes the following source data and figure supplement(s) for figure 1:

**Source data 1.** Quantification of overlap between DA8 and DA9 synaptic domains.

**Figure supplement 1.** Colocalization of GFP::RAB-3 and mCherry::RAB-3 in the DA9 neuron with two synaptic tiling markers.

**Figure supplement 1—source data 1.** Fluorescence intensity of GFP and mCherry channels.

**Figure supplement 2.** P*mig-13::mCherry::rab-3* is not ectopically expressed in DA8 of *unc-4(e120)* mutants.

**Figure supplement 3.** *unc-4* and *unc-37* may function in the same genetic pathway as *plx-1*.

**Figure supplement 3—source data 1.** Quantification of overlap between DA8 and DA9 synaptic domains in *plx-1* mutant animals.

**Figure supplement 4.** *plx-1* and *rap-2* expression is largely unaffected in *unc-4* and *unc-37* mutants.

**Figure supplement 4—source data 1.** Quantification of animals that expressed P*plx-1::GFP* in the anterior and posterior DA8/DA9 of wild-type, *unc-4 (e120)*, and *unc-37(e262)* animals.

**Figure supplement 4—source data 2.** Quantification of animals (n=100) that expressed P*rap-2::GFP* in the anterior and posterior DA8/9 of wild-type, *unc-4*, and *unc-37* animals.

signaling components. Surprisingly, temporal knockdown of *unc-4* using *unc-4* temperature-sensitive mutants and the auxin-inducible degron (AID) system showed that *unc-4* is required for synaptic tiling in the postmitotic DA neurons, but not required during the development of DA neurons. On the other hand, *unc-37* is required in both the developing DA neurons and the postmitotic DA neurons. Interestingly, the expression patterns of A-type and B-type cholinergic neuronal cell fate markers are largely unaffected in the DA8 and DA9 neurons of *unc-4* and *unc-37* mutants, suggesting that the synaptic tiling defects in *unc-4* and *unc-37* are less likely due to the secondary defects of the cell fate determination. We also found that the Wnt-induced ectopic expression of *ceh-12* homeobox gene underlies, at least in part, the synaptic tiling defects of *unc-4* mutants. Taken together, we provide the postmitotic roles of cell fate determinants in precise synapse pattern formation.

## Results

### *unc-4* homeobox gene and *unc-37/Groucho* are required for the synaptic patterning of DA8 and DA9 neurons

The DA neurons extend axons within the dorsal nerve cord, where they form *en passant* synapses at specific subaxonal regions onto the dorsal body wall muscles (*White et al., 1976*). Previous electron microscopy reconstruction and fluorescent microscopy has revealed the tiled synaptic innervation pattern between neighboring DA neurons: the axonal region with synapses (synaptic domain) of a DA neuron does not overlap with those of its neighboring DA neurons (*Chen et al., 2018*; *Mizumoto and Shen, 2013a*; *White et al., 1976*). In this study we generated a new transgene (*mizIs3*) with improved color contrast between the two most posterior DA neurons, DA8 and DA9, under the fluorescent microscope. *mizIs3* expresses the synaptic vesicle-associated protein, RAB-3, fused with a brighter GFP variant (GFPnovo2) (*Hendi and Mizumoto, 2018*) and ZF1 degron sequences (*Armenti et al., 2014*) in all DA neurons. It also expresses mCherry::RAB-3 and ZIF-1 in DA9. The expression of ZIF-1 in the DA9 neuron results in the ubiquitin-mediated protein degradation of ZF1::GFPnovo2::RAB-3, thereby allowing DA8 and DA9 synapses to be labeled in green and red, respectively (see experimental procedures) (*Figure 1A*). As previously described, DA8 and DA9 exhibited tiled synaptic innervation with minimal overlap in wild-type animals (*Figure 1B*; *Chen et al., 2018*; *Mizumoto and Shen, 2013a*).

In *unc-4(e120)* null mutants, we found significant overlap between the DA8 and DA9 synaptic domains: the synapses of DA8 and DA9 are intermingled in the region of overlap (*Figure 1C,E*). We noticed that the degradation of ZF1::GFPnovo2::RAB-3 was incomplete and mCherry::RAB-3 expression was slightly weaker in DA9 neurons of *unc-4* mutants (*Figure 1C*). This is likely due to the reduced activity of the *mig-13* promoter used to express ZIF-1 and mCherry::RAB-3 (see below). Indeed, the synaptic pattern of DA8 and DA9 in *unc-4(e120); mizIs3* resembled that of *unc-4(e120); wyIs446*, which carries another synaptic tiling marker (*wyIs446*) we used in our previous work (*Chen et al., 2018*). *wyIs446* transgene does not use the ZIF-1/ZF1 degradation system, and DA9 synapses are labeled with both GFP- and mCherry-fused RAB-3 (*Figure 1—figure supplement 1A*). In both synaptic tiling markers, mCherry::RAB-3 puncta colocalized with green fluorescent signal in *unc-4* mutants (*Figure 1—figure supplement 1B,C*). Therefore, the synaptic tiling defects of *unc-4* is not due to the compromised ZIF-1/ZF1 degron system. To exclude the possibility that the anteriorly extended DA9 synaptic domain judged by the mCherry RAB-3 in *unc-4* mutants is due to the misexpression of mCherry::RAB-3 in DA8, we examined the mCherry::RAB-3 signal in the cell body of DA8 and DA9. In wild type, the dim mCherry::RAB-3 signal is observed in the cell body of DA9, which can only be visible with increased image brightness (*Figure 1—figure supplement 2A*). This mCherry signal is absent in the DA8 cell body in both wild-type and *unc-4* mutants (*Figure 1—figure supplement 2A,B*). We therefore concluded that the mCherry::RAB-3 puncta represent DA9 synapses, and the synaptic tiling defect in *unc-4* mutants is not due to the ectopic expression of mCherry::RAB-3 in DA8.

Similar to *unc-4*, hypomorphic mutants of *unc-37(e262)*, which carry a single amino acid substitution (H539Y) (*Pflugrad et al., 1997*), exhibited significant overlap between the DA8 and DA9 synaptic domains, indicating that *unc-37* is also required for synaptic tiling (*Figure 1D,E*). Moreover, *unc-4 (e120)* did not enhance the synaptic tiling defects of *unc-37(e262)* (*Figure 1E*). This result is consistent with the fact that *unc-4* and *unc-37* function as a transcriptional repressor complex in the A-type

motor neurons (*Winnier et al., 1999*). Approximately 15% of *unc-4* and *unc-37* mutants showed subtle defasciculation phenotype of the DA8 and DA9 axons (data not shown). As physical contact between the DA8 and DA9 axons is required for synaptic tiling (*Mizumoto and Shen, 2013a*), we excluded these animals from our quantification.

UNC-4 physically interacts with UNC-37 via its carboxy-terminal Engrailed-like repressor (eh1) domain (*Winnier et al., 1999*). To further test whether *unc-4* and *unc-37* function together in tiled synaptic innervation, we examined the synaptic tiling in *unc-4(e26)* mutants which carry a missense mutation in the eh1 domain that has been shown to disrupt the physical interaction between UNC-4 and UNC-37 (*Winnier et al., 1999*). Similar to *unc-4(e120)* null mutants, we observed severe synaptic tiling defects between the DA8 and DA9 neurons in *unc-4(e26)* mutant animals (*Figure 1E*). This result supports the idea that UNC-4 and UNC-37 act together as a repressor complex for proper tiled synaptic innervation.

## *unc-4* and *unc-37* may function in the same genetic pathway as *plx-1/ Plexin*

We have previously shown that Semaphorin (Sema)–Plexin signaling controls the synaptic tiling of DA8 and DA9 (*Chen et al., 2018*; *Mizumoto and Shen, 2013a*). In the loss of function mutants of *smp-1/Sema*, *plx-1/Plexin*, and *rap-2/Rap2A*, DA8 and DA9 show severe synaptic tiling defects (*Chen et al., 2018*; *Mizumoto and Shen, 2013a*). Since UNC-4 and UNC-37 are transcriptional regulators, it is possible that they regulate synaptic tiling by controlling the expression of Sema–Plexin signaling components. Consistent with this idea, *plx-1* mutants did not enhance the synaptic tiling defects of *unc-4* or *unc-37* mutants (*Figure 1—figure supplement 3A–D*). This result suggests that both *unc-4* and *unc-37* may function in the same genetic pathway as *plx-1*. However, we do not exclude the possibility that synaptic tiling defects of *unc-4* and *unc-37* mutants are at the maximum severities so that *plx-1* cannot enhance the phenotype.

We have previously shown that PLX-1::GFP is localized at the anterior edge of the DA9 synaptic domain where it locally inhibits synapse formation in a *smp-1*-dependent manner (*Figure 1—figure supplement 3E*; *Mizumoto and Shen, 2013a*). In *unc-37* mutants, PLX-1::GFP localization was unaffected (*Figure 1—figure supplement 3G*), suggesting that *smp-1* functions properly in the absence of *unc-37*. The expression of PLX-1::GFP in *unc-4(e120)* mutants was too dim to examine its subcellular localization in a reliable manner due to the diminished *itr-1* promoter activity, which we used to express PLX-1::GFP (*Figure 1—figure supplement 3H*). In a temperature-sensitive mutant of *unc-4 (e2322ts)* (*Miller et al., 1992*; *Winnier et al., 1999*), which show severe synaptic tiling defects when grown at the restrictive temperature (25°C) (see below), the expression from the *itr-1* promoter was strong enough to examine the PLX-1::GFP localization. We observed normal PLX-1::GFP localization in the DA9 axon (*Figure 1—figure supplement 3F*), suggesting that *smp-1* also functions properly in *unc-4* mutants. We then tested the possible functions of *unc-4* and *unc-37* on *plx-1* expression using a P*plx-1::GFP* transcription reporter and found a minor reduction in *plx-1* expression in *unc-37* mutants (*Figure 1—figure supplement 4A–D*). However, this subtle reduction in *plx-1* expression is unlikely to explain the complete penetrant synaptic tiling defect of *unc-4* and *unc-37* mutants. Furthermore, expression of *plx-1* cDNA under the DA-neuron-specific promoter (P*unc-4c*) did not rescue the synaptic tiling defects of *unc-4* or *unc-37* mutants (data not shown).

While PLX-1::GFP localization was not affected in *unc-4* and *unc-37* mutants, we observed the ectopic synapse formation anterior to the PLX-1::GFP domain (*Figure 1—figure supplement 3F and G*). This phenotype is reminiscent to the mutant phenotype of PLX-1 downstream effectors such as *rap-2* and *mig-15*, in which PLX-1 cannot locally inhibit synapse formation (*Chen et al., 2018*). However, we did not observe a reduction in *rap-2* expression in *unc-4* and *unc-37* mutants (*Figure 1—figure supplement 4E–H*), and the expression of known Sema/Plexin signaling components (*rap-2* and *mig-15*) under the DA neuron-specific promoter (P*unc-4c*) did not rescue the synaptic tiling defects of *unc-4* and *unc-37* mutants (data not shown). It is therefore possible that *unc-4* and *unc-37* regulate synaptic tiling by controlling the expression of previously uncharacterized Sema–Plexin signaling components.

## DA8 and DA9 cell fates are largely unaffected in *unc-4* and *unc-37* mutants

As *unc-4* and *unc-37* determine the A-type motor neuron fates by repressing the expression of B-type motor neuron-specific genes (*Kerk et al., 2017*; *Winnier et al., 1999*), the synaptic tiling defects of *unc-4* and *unc-37* mutants may be a secondary defect of the abnormal neuronal cell fates in DA8 and DA9 neurons. We therefore examined the neuronal identity of DA8 and DA9 using a DA cell fate marker, P*unc-53* (*Kerk et al., 2017*), and three DB cell fate markers, P*unc-129DB*, P*acr-16*, and P*acr-5* (*Kerk et al., 2017*; *Winnier et al., 1999*) under the fluorescent microscope. We co-expressed *his-24* fused to mCherry under the *unc-4c* promoter to label DA8 and DA9 nuclei. As relative position of the DA8 and DA9 cell bodies was variable, we distinguished them as anterior DA8/9 or posterior DA8/9 for examining these DA and DB cell fate markers. In wild-type animals, both anterior and posterior DA8/DA9 expressed the DA marker, P*unc-53* but not the DB markers, P*acr-16*, P*unc-129DB*, and P*acr-5* (*Table 1*). In the *unc-4(e120)* and *unc-37(e262)* mutant animals, the expression of P*unc-53* was not affected in both anterior and posterior DA8/9 neurons, suggesting that DA8 and DA9 retain A-type identity (*Table 1*). The expression of a DB marker P*unc-129DB* in DA8/9 of *unc-4* and *unc-37* mutants was not different from wild type, while there was a slight increase in the number of animals expressing the other DB markers, P*acr-16* and P*acr-5*, with a slightly higher penetrance in the posterior DA8/9 (*Table 1*). These results suggest that DA8 and DA9 might have minor cell fate defects that result in the leaky expression of some DB markers. However, the low penetrance of the partial cell fate defects is less likely to account for the fully penetrant synaptic tiling defects of *unc-4* and *unc-37* mutants.

We further tested the identity of DA9 neurons using DA9-specific markers, P*mig-13* (*mizIs3*), P*glr-4* (*otIs476*), and P*itr-1* (*wyIs320*) (*Klassen and Shen, 2007*; *Kratsios et al., 2017*). These DA9 markers label axon trajectory and/or synaptic varicosity allowing us to distinguish the DA9 neuron from the DA8 neuron. These DA9 markers were expressed in DA9 of wild-type animals at full penetrance (*Table 1*). In all *unc-4* mutants, we observed expression of P*glr-4* and P*mig-13*, while P*mig-13* expression was noticeably weaker than wild type as described above (*Table 1* and *Figure 1C*). The P*itr-1* expression was also weaker in *unc-4* mutants, and we were unable to reliably examine the dim expression of the P*itr-1* marker in *unc-4* mutants under the fluorescent microscope due to the high expression of the P*itr-1* in the hindgut adjacent to the DA9 cell body (*Table 1*, *Figure 1—figure supplement 3H*). However, P*itr-1* expression was detected in DA9 at full penetrance under the confocal microscope (n = 40). In *unc-37* mutant animals, the expression of these DA9 specific markers was mostly unaffected, with a small fraction of animals (7/100) lacking P*glr-4* expression in DA9 (*Table 1*).

Together we conclude that despite the subtle changes in the expression of some of the cell fate markers, the cell fates of DA8 and DA9 in *unc-4* and *unc-37* mutants are largely unaffected, and therefore, it is unlikely that the synaptic tiling defects of *unc-4* and *unc-37* mutants are due to the secondary defects of cell fate transformation of the DA neurons.

## *unc-4* and *unc-37* are necessary and sufficient in the DA neurons to regulate synaptic tiling

We next determined in which cells *unc-4* and *unc-37* function by using the AID system to conditionally control the degradation of UNC-4 and UNC-37 in a spatiotemporally controlled manner. A plant F-box protein, TIR1, mediates the degradation of AID-tagged proteins in an auxin-dependent manner (*Nishimura et al., 2009*; *Zhang et al., 2015*). Using CRISPR/Cas9-mediated genome editing, we

**Table 1.** Expression of cell fate markers in the anterior or posterior DA8/DA9 neurons.

| Genotype | DA marker | | DB markers | | | | | | DA9 markers | | |
|---|---|---|---|---|---|---|---|---|---|---|---|
| | P*unc-53* | | P*unc-129DB* | | P*acr-16* | | P*acr-5* | | P*mig-13* | P*glr-4* | P*itr-1* |
| | Anterior | Posterior | Anterior | Posterior | Anterior | Posterior | Anterior | Posterior | DA9 | DA9 | DA9 |
| Wild type | 94/100 | 94/100 | 2/100 | 4/100 | 0/100 | 0/100 | 0/100 | 0/100 | 100/100 | 100/100 | 100/100 |
| *unc-4(e120)* | 94/100 | 91/100 | 3/100 | 4/100 | 3/100 | 8/100 | 4/100 | 8/100 | 100/100 | 99/100 | n.d. |
| *unc-37(e262)* | 92/100 | 92/100 | 1/100 | 8/100 | 2/100 | 24/100 | 1/100 | 30/100 | 100/100 | 93/100 | 98/100 |

tagged endogenous *unc-4* and *unc-37* with AID and blue fluorescent protein (BFP) at the 3' end of the *unc-4* locus (*miz40[unc-4::AID::BFP]*) and *unc-37(miz36[unc-37::AID::BFP])* (*Figure 2—figure supplement 1B*). Consistent with previous studies, we observed sustained expression of the UNC-4::AID::BFP and UNC-37::AID::BFP fusion proteins in the nuclei of DA and VA neurons (*Fox et al., 2005*; *Miller and Niemeyer, 1995*; *Pflugrad et al., 1997*) at the L4 stage (*Figure 2—figure supplement 2A,C*). We then crossed *unc-4(miz40)* and *unc-37(miz36)* with a strain carrying *mizSi3* (P*unc-4c::TIR1*), a transgene that expresses TIR1 specifically in the DA neurons under the DA neuron-specific promoter (P*unc-4c*) (*Figure 2—figure supplement 1A*). *unc-4(miz40); mizSi3* and *unc-37(miz36); mizSi3* animals did not exhibit uncoordinated locomotion phenotype (data not shown) or synaptic tiling defects in the absence of auxin (*Figures 2A, C, and E*), suggesting that the UNC-4::AID::BFP and UNC-37::AID::BFP fusion proteins are fully functional. When *unc-4(miz40); mizSi3* and *unc-37 (miz36); mizSi3* animals were grown in the presence of the water-soluble synthetic auxin analog (K-NAA) (*Martinez et al., 2020*), we observed loss of UNC-4::AID::BFP and UNC-37::AID::BFP signal in DA neurons, but not in VA neurons, suggesting the successful DA neuron-specific degradation of UNC-4 and UNC-37 (*Figure 2—figure supplement 2B,D*). Furthermore, *unc-4(miz40); mizSi3* and *unc-37(miz36); mizSi3* animals grown on K-NAA-containing plates did not exhibit backwards locomotion defects, which is attributed to the AVA>VA wiring defects in *unc-4* and *unc-37* mutants, suggesting UNC-4 and UNC-37 are fully functional in the VA neurons during DA-specific degradation of UNC-4 and UNC-37. Using this system, we first conducted the DA-specific continuous degradation of UNC-4 and UNC-37 by placing the L4 parental animals on K-NAA-containing plates and observing the progeny at the L4 stage. We observed severe synaptic tiling defects between the DA8 and DA9 synaptic domains (*Figure 2B, D, and E*), suggesting that *unc-4* and *unc-37* are required in the DA neurons for proper synaptic tiling.

We also performed tissue-specific rescue experiments by expressing the *unc-4* and *unc-37* cDNAs under the DA-specific *unc-4c* promoter in *unc-4(e120)* and *unc-37(e262)* mutant animals, respectively. We co-expressed *his-24* fused to *GFPnovo2*, under the *unc-4c* promoter to determine which DA neurons carry the extrachromosomal array. We observed significant rescue of the synaptic tiling defect when *unc-4* is expressed in both DA8 and DA9 neurons of *unc-4* mutant animals (*Figure 2F*). Similarly, we observed significant rescue of the synaptic tiling defects in *unc-37* mutant animals when *unc-37* is expressed in both DA8 and DA9 neurons (*Figure 2G*). However, in the genetic mosaic animals in which *unc-4* or *unc-37* is expressed only in DA9, we did not observe rescue of the synaptic tiling defects (*Figure 2F,G*). This suggests that *unc-4* and *unc-37* are not sufficient in DA9 and instead function in both DA8 and DA9 neurons to control synaptic tiling.

## *unc-37* but not *unc-4* is required during embryonic development for proper synaptic tiling

The lack of apparent cell fate defects in DA8 and DA9 of *unc-4* and *unc-37* mutants suggests that *unc-4* and *unc-37* control tiled synaptic innervation independent of the cell fate determination function of the A-type motor neurons. We therefore sought to determine the temporal requirement of *unc-4* and *unc-37* in synaptic tiling. The DA neurons are born in the late gastrulation stage and completes the development during late embryogenesis (*Sulston et al., 1983*). The tiled synaptic innervation of DA8 and DA9 is observed at the early first larval (L1) stage in wild-type animals (*Mizumoto and Shen, 2013a*). To test the temporal requirement of *unc-4* in tiled synaptic patterning, we used a reversible temperature-sensitive allele of *unc-4* (*Miller et al., 1992*; *Winnier et al., 1999*). *unc-4(e2322ts)* mutants carry a point mutation (L121F) that resides in the homeodomain (*Winnier et al., 1999*). The *unc-4(e2322ts)* mutation is permissive at 16°C and restrictive at 25°C (*Miller et al., 1992*). When grown at the permissive temperature (16°C), *unc-4(e2322ts)* mutants did not exhibit synaptic tiling defects compared with wild type (*Figure 3B,F*). When grown at the restrictive temperature (25°C), *unc-4(e2322ts)* mutants showed severe synaptic tiling defects (*Figure 3C,F*). We then determined if *unc-4* functions during DA neurogenesis or in postmitotic DA neurons to regulate synaptic tiling by down-shifting or up-shifting the growth temperature at the early L1 stage (*Figure 3A*). Interestingly, we did not observe significant synaptic tiling defects when *unc-4* was knocked down embryonically by down-shifting the growth temperature of *unc-4(e2322ts)* mutant animals from 25°C to 16°C at the L1 stage (*Figure 3D,F*). This result indicates that *unc-4* is not required for synaptic tiling during the embryonic stage when the DA8 and DA9 cell fates are determined by *unc-4*. Conversely, the postembryonic *unc-4* knockdown by up-shifting the growth

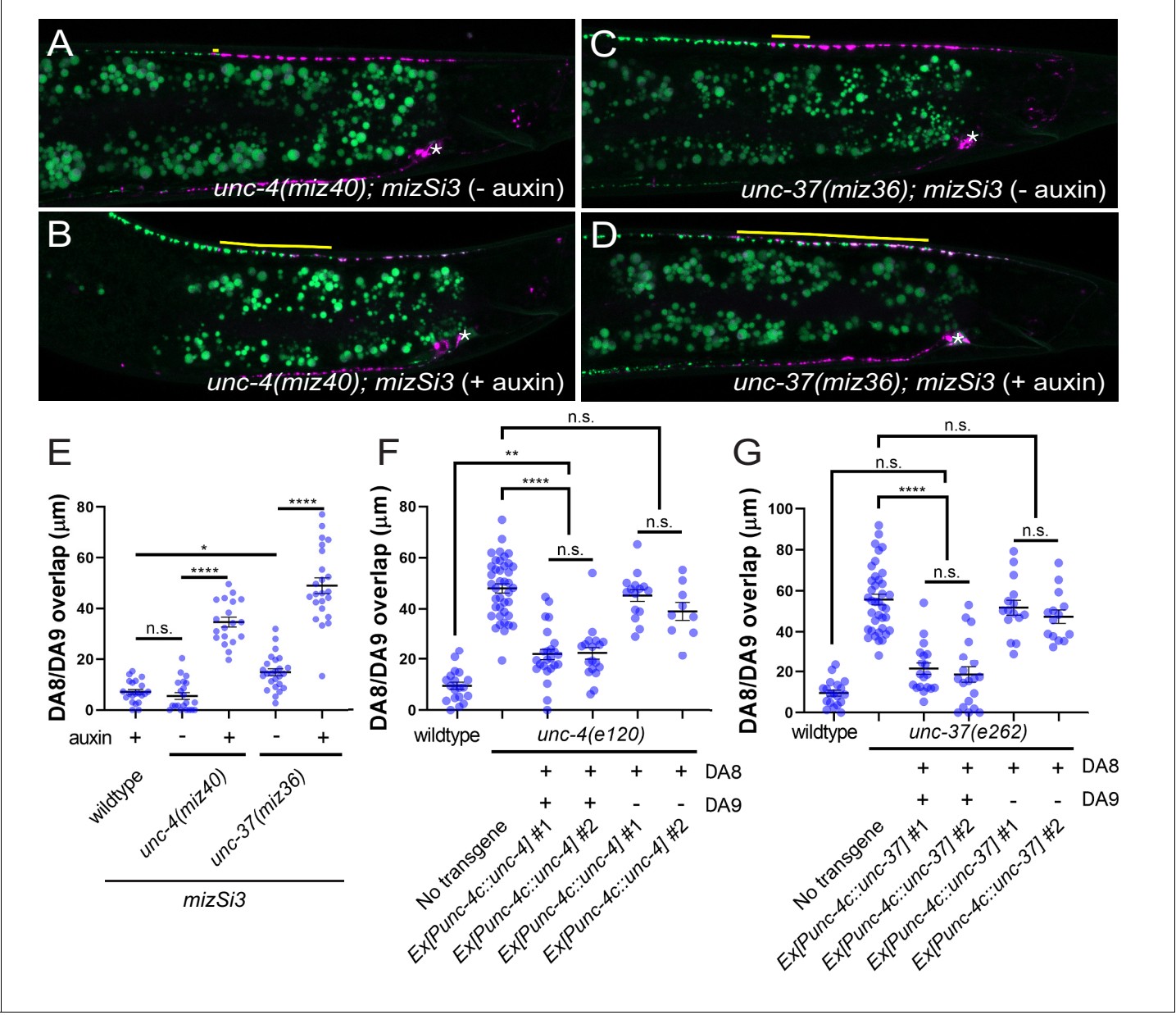

**Figure 2.** *unc-4* and *unc-37* are necessary and sufficient for the tiled innervation of DA8 and DA9 neurons. (A–D) Representative images of synaptic tiling in *unc-4(miz40); mizSi3 (− auxin)* (A), *unc-4(miz40); mizSi3 (+ auxin)* (B), *unc-37(miz36); mizSi3 (− auxin)* (C), and *unc-37(miz36); mizSi3 (+auxin)* (D). *mizSi3* expresses TIR1 in the DA neurons for tissue-specific degradation of UNC-4 and UNC-37. The overlap between the DA8 and DA9 synaptic domains are highlighted with yellow lines. Asterisks: DA9 cell body. Scale bar: 10 μm. (E) Quantification of overlap of DA8 and DA9 synaptic domains of *unc-4(miz40); mizSi3* and *unc-37(miz36); mizSi3* animals treated with (+) and without (-) auxin. See *Figure 2—source data 1*. (F, G) Cell specific rescue of *unc-4(e120)* mutants with *unc-4* cDNA using DA-specific promoter (P*unc-4c*). See *Figure 2—source data 2*. (F) Cell specific rescue of *unc-37(e262)* mutants with *unc-37* cDNA using DA-specific promoter (P*unc-4c*). See *Figure 2—source data 3*. (G). Two independent transgenic lines were quantified. Mosaic experiment using P*unc-4c::unc-4* and P*unc-4c::unc-37* are also shown. Each dot represents a single animal. Black bars indicate mean ± SEM. n.s.: not significant; *: p<0.05; **: p<0.01; ****:p<0.0001.

The online version of this article includes the following source data and figure supplement(s) for figure 2:

**Source data 1.** Quantification of overlap of DA8 and DA9 synaptic domains of *unc-4(miz40); mizSi3* and *unc-37(miz36); mizSi3* animals treated with (+) and without (-) auxin.

**Source data 2.** Cell specific rescue of *unc-4(e120)* mutants with *unc-4* cDNA using DA-specific promoter (P*unc-4c*).

**Source data 3.** Cell specific rescue of *unc-37(e262)* mutants with *unc-37* cDNA using DA-specific promoter (P*unc-4c*).

**Figure supplement 1.** Genomic structure of the AID strains.

**Figure supplement 2.** DA-neuron-specific degradation of UNC-4 and UNC-37 using AID system.

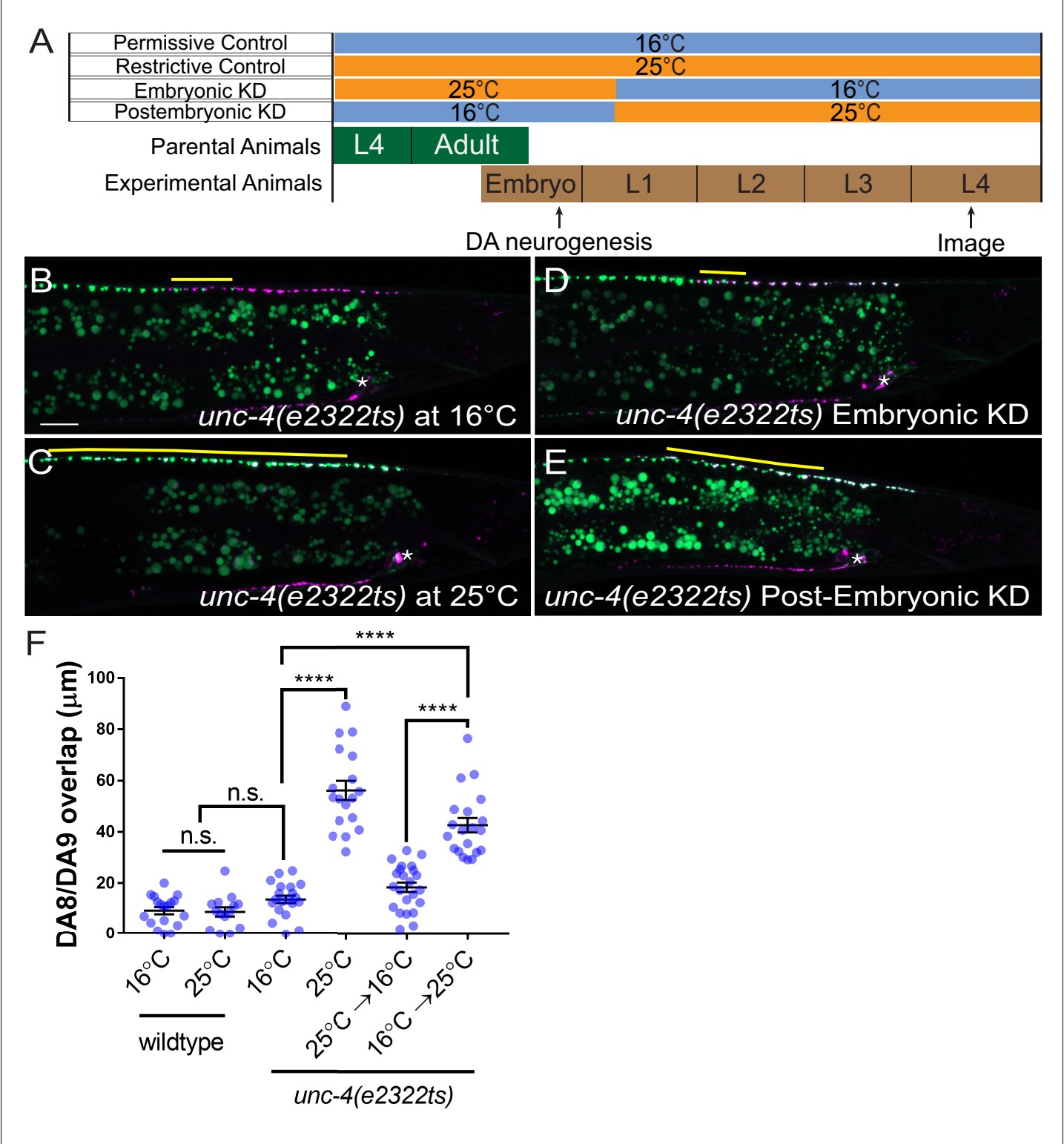

**Figure 3.** *unc-4* functions in postmitotic DA neurons for the synaptic tiling of DA8 and DA9 neurons. (**A**) Experimental design of the temperature shift assay. (**B–E**) Representative images of synaptic tiling in *unc-4(e2322ts)* at 16°C (**B**), *unc-4(e2322ts)* at 25°C (**C**), *unc-4(e2322ts)* embryonic knockdown (**D**), and *unc-4(e2322ts)* postembryonic knockdown (**E**). The overlap between the DA8 and DA9 synaptic domains are highlighted with yellow lines. Asterisks: DA9 cell body. Scale bar: 10 μm. (**F**) Quantification of overlap of DA8 and DA9 synaptic domains. See *Figure 3—source data 1*. Each dot represents a single animal. Black bars indicate mean ± SEM. n.s.: not significant; ****:p<0.0001.

The online version of this article includes the following source data for figure 3:

**Source data 1.** Quantification of overlap of DA8 and DA9 synaptic domains.

temperature from 16°C to 25°C at the L1 stage resulted in severe synaptic tiling defects (*Figure 3E, F*). Together, this suggests that *unc-4* is required in the postmitotic DA8 and DA9 neurons in which the cell fates are already set.

To verify the role of *unc-4* and *unc-37* in the postmitotic DA neurons, we conducted DA neuron-specific temporal degradation experiments using the AID system described above. We first conducted UNC-4 and UNC-37 degradation in DA neurons specifically during embryonic development. The L4 parental animals of *unc-4(miz40); mizSi3* and *unc-37(miz36); mizSi3* were cultured on K-NAA containing plates, and their progeny were transferred at the early L1 stage to the control plates (*Figure 4A*). Consistent with the temperature shift experiments using *unc-4(e2322ts)* mutants, we did not observe any synaptic tiling defects when UNC-4 was degraded during embryonic development (*Figure 4B,F*). On the other hand, embryonic degradation of UNC-37 resulted in severe synaptic tiling defects (*Figure 4C,F*), suggesting that UNC-37 but not UNC-4 is required during embryonic development to regulate synaptic tiling.

We have previously shown that synaptic tiling between DA8 and DA9 is observed as early as L1 stage (*Mizumoto and Shen, 2013a*). To determine if *unc-4* and *unc-37* are required for establishing the synaptic tiling, we examined the effect of embryonic degradation of UNC-4 and UNC-37 on synaptic tiling at early L2 stage. We found that embryonic degradation of UNC-37 but not UNC-4 caused synaptic tiling defects (*Figures 4D, E, and G*). Our results suggest that *unc-4* is not required to establish the synaptic tiling, whereas *unc-37* may be required to establish the synaptic tiling during embryonic development.

## *unc-4* and *unc-37* are required throughout larval development in postmitotic DA neurons to regulate synaptic tiling

Next, we examined if *unc-4* and *unc-37* are required in the postmitotic DA neurons to regulate synaptic tiling using the AID system. We first transferred early L1 animals from the control plates to the K-NAA-containing plates and let them grow until the L4 stage (*Figure 5A*). Consistent with our findings using the temperature-sensitive mutant of *unc-4*, we found that postembryonic degradation of UNC-4 resulted in synaptic tiling defects, which were comparable to the continuous UNC-4 degradation (*Figure 5B,F*). Together with the lack of synaptic tiling defects after embryonic degradation of UNC-4, these data suggest that *unc-4* is exclusively required in the postmitotic DA neurons to regulate synaptic tiling. We then examined the effect of postembryonic degradation of UNC-37 on synaptic tiling. Similar to *unc-4*, postembryonic degradation of UNC-37 resulted in severe synaptic tiling defects (*Figure 5C,F*). This suggests that, unlike *unc-4*, *unc-37* is required for the synaptic tiling during DA neurogenesis as well as in the postmitotic DA neurons. Given that *unc-4* is specifically required in the postmitotic DA neurons, this result suggests that *unc-37* may function with other transcription factors during DA neurogenesis for synaptic tiling (Figure 7).

To determine whether *unc-4* and *unc-37* are required for the maintenance of the synaptic tiling in the postmitotic DA neurons, we conducted postembryonic degradation of UNC-4 and UNC-37 at a later larval stage (mid-L2) and examined the synaptic phenotype at the L4 stage (*Figure 5A*). We found that degradation of UNC-4 and UNC-37 after mid-L2 stage caused mild yet significant synaptic tiling defects (*Figure 5D–F*). Weaker synaptic tiling defects in the late postembryonic degradation of UNC-4 and UNC-37 suggest that *unc-4* and *unc-37* are required throughout postembryonic development for the maintenance of the synaptic tiling. Our results are consistent with the potential roles of *unc-4* during L2 and L3 stages in VA neurons to maintain the synaptic wiring of the VA neurons (*Miller et al., 1992*).

## *ceh-12* functions downstream of *unc-4* and *unc-37* in synaptic tiling

It has been shown that *unc-4* and *unc-37* repress the expression of *ceh-12,* a member of the HB9 family of Homeobox genes, in the VA motor neurons to specify the AVA>VA synaptic connections (*Von Stetina et al., 2007*). Loss of *ceh-12* partially restores the synaptic input and locomotion defects of *unc-4* and *unc-37* mutants (*Von Stetina et al., 2007*). Furthermore, the ectopic expression of *ceh-12* in VA neurons is sufficient to induce *unc-4* mutant-like uncoordinated locomotion phenotype (*Von Stetina et al., 2007*). To determine if *ceh-12* also functions downstream of *unc-4* and *unc-37* in regulating tiled synaptic innervation, we first examined the *ceh-12* expression in DA8 and DA9 neurons using a transcriptional reporter transgene *wdIs62 (Pceh-12::GFP)*. P*ceh-12::GFP* expression

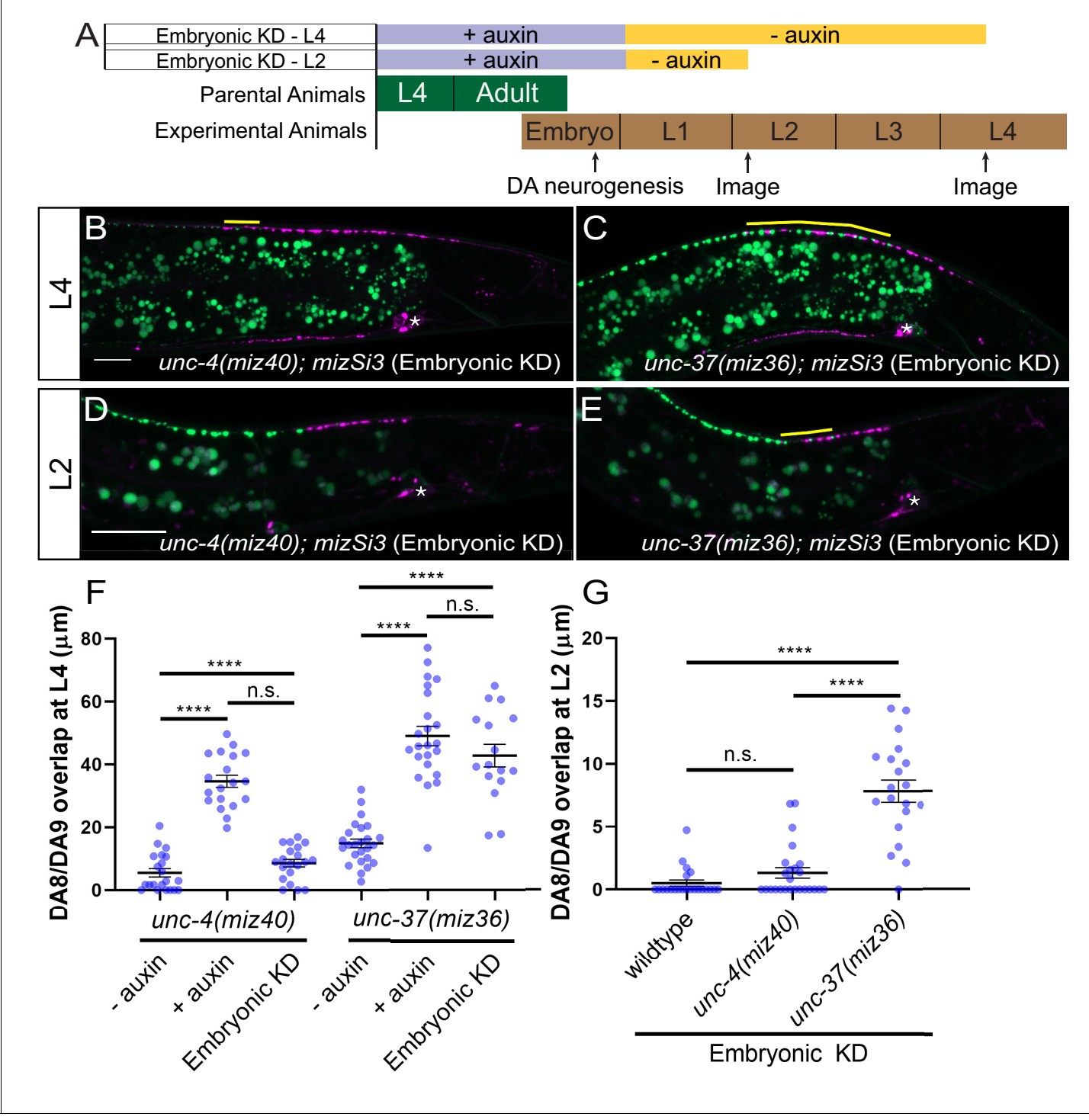

**Figure 4.** *unc-37* but not *unc-4* is required during DA neurogenesis for proper synaptic tiling of DA8 and DA9 neurons. (A) Experimental design of the embryonic degradation of UNC-4 and UNC-37 using the AID system. (B, C) Representative images of synaptic tiling at L4 stage after embryonic degradation in *unc-4(miz40); mizSi3* (B) and *unc-37(miz36); mizSi3* (C). (D, E) Representative images of synaptic tiling at L2 stage after embryonic degradation in *unc-4(miz40); mizSi3* (D) and *unc-37(miz36); mizSi3* (E). The overlap between the DA8 and DA9 synaptic domains are highlighted with yellow lines. Asterisks: DA9 cell body. Scale bar: 10 μm. (F) Quantification of overlap of DA8 and DA9 synaptic domains at the L4 stage after embryonic auxin treatment. See *Figure 4—source data 1*. (G) Quantification of overlap of DA8 and DA9 synaptic domains at the L2 stage after embryonic auxin treatment. See *Figure 4—source data 2*. Each dot represents a single animal. Black bars indicate mean ± SEM. n.s.: not significant; ****:p<0.0001. The online version of this article includes the following source data for figure 4:

**Source data 1.** Quantification of overlap of DA8 and DA9 synaptic domains at the L4 stage after embryonic auxin treatment.

**Source data 2.** Quantification of overlap of DA8 and DA9 synaptic domains at the L2 stage after embryonic auxin treatment.

was not detected in DA8 and DA9 of wild type at L1, L2, or L4 stage (*Figure 6—figure supplement 1A–A″*). We observed weak ectopic expression of P*ceh-12::GFP* in DA8 and DA9 neurons at L2 and L4 stage in *unc-4* and *unc-37* mutants (*Figure 6—figure supplement 1B′–B″ and C′-C′*). However, at the L1 stage, the ectopic P*ceh-12::GFP* expression in DA8 and DA9 was only detected in *unc-37* but not in *unc-4* mutants (*Figure 6—figure supplement 1B and C*). This observation is consistent with our temporal knockdown experiments of *unc-4* and *unc-37* which demonstrated that the function of *unc-4* is exclusive in the postembryonic DA neurons while *unc-37* functions in both embryonic and postembryonic DA neurons.

To test whether the ectopic expression of *ceh-12* in the DA neurons underlies the synaptic tiling defects of *unc-4* and *unc-37*, we expressed *ceh-12* cDNA using the DA neuron-specific promoter (P*unc-4c*) in wild-type animals. We observed significant synaptic tiling defects in these animals (*Figure 6A,C*), suggesting that *unc-4* and *unc-37* likely repress the expression of *ceh-12* in DA neurons for proper synaptic tiling. To exclude the possibility that the synaptic tiling defects of the *ceh-12*-expressing animals are due to the ectopic expression of cell fate determinants in general, we expressed *unc-55* in the DA neurons. *unc-55* is expressed in VD GABAergic motor neurons and is required for repressing DD neuron-specific gene expression (*Shan et al., 2005*; *Zhou and Walthall, 1998*). Ectopic expression of *unc-55* in DA neurons did not cause synaptic tiling defects (*Figure 6B, C*). While these results strongly suggest that *ceh-12* functions downstream of *unc-4* and *unc-37* in synaptic tiling, *ceh-12* mutation did not suppress the synaptic tiling defects of *unc-4* and *unc-37* mutants (*Figure 6D*). This suggests that the weak ectopic expression of *ceh-12* in *unc-4* and *unc-37* mutants is not sufficient to cause the synaptic tiling defects, and there are additional factor(s) that function redundantly with *ceh-12* under the control of *unc-4* and *unc-37* in synaptic tiling (*Figure 7*).

## *bar-1/β-catenin* partially suppresses the synaptic tiling defect of *unc-4* and *unc-37* mutants

Previous work has shown that EGL-20, a Wnt morphogen that is expressed in the cells around the preanal ganglion (*Whangbo and Kenyon, 1999*), acts through the canonical Wnt signaling pathway components including *bar-1/β-catenin*, to positively regulate the expression of *ceh-12* in VA motor neurons (*Schneider et al., 2012*). *unc-4* and *unc-37* inhibits the expression of *ceh-12* by repressing the expression of the Frizzled receptors, *mom-5* and *mig-1* (*Schneider et al., 2012*). Given that the canonical EGL-20/Wnt signaling acts as a positive regulator of *ceh-12* expression, it is possible that Wnt signaling induces the expression of other transcription factors that may function redundantly with *ceh-12* in synaptic tiling (*Figure 7*). To test this possibility, we examined the synaptic tiling defects in *unc-4(e120); bar-1(ga80)* and *unc-37(e262); bar-1(ga80)* double mutants. Consistent with our idea, *bar-1(ga80)* partially suppressed the synaptic tiling defect of *unc-4* and *unc-37* mutants (*Figure 6E*). This result suggests that UNC-4 and UNC-37 control synaptic tiling by inhibiting the canonical Wnt signaling for the expression of multiple genes including *ceh-12*.

## Discussion

In this study, we showed that the cell fate determinants of A-type cholinergic motor neurons, UNC-4 homeobox transcription factor and its corepressor UNC-37/Groucho, have novel roles in the postmitotic neurons to control the tiled presynaptic innervation pattern. Previous studies in both vertebrate and invertebrate systems have demonstrated that sustained expression of neuronal cell fate determinants in the differentiated neurons is required to maintain effector gene expression for the functionality of neurons (*Altun-Gultekin et al., 2001*; *Carney et al., 2013*; *Cheng et al., 2004*). For example, sustained expression of Nurr1 is required for the survival of the midbrain dopaminergic neurons and maintained expression of the genes required for dopamine synthesis and transport (*Kadkhodaei et al., 2009*; *Saucedo-Cardenas et al., 1998*; *Zetterström et al., 1997*). Our temporal knockdown experiments using temperature-sensitive mutants and the AID system revealed the critical functions of sustained expression of *unc-4* and *unc-37* in synapse pattern formation. Our work

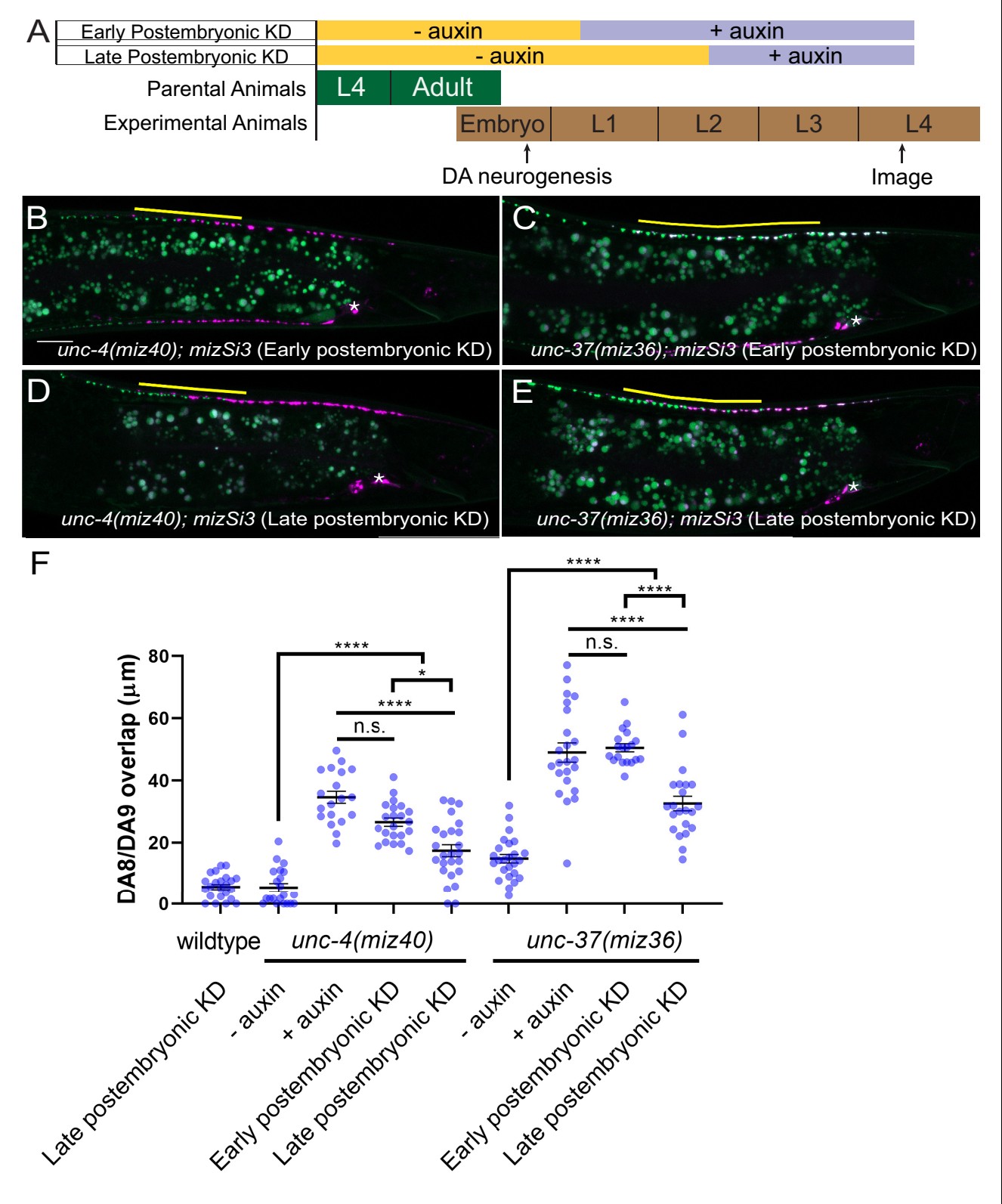

**Figure 5.** *unc-4* and *unc-37* are required in the postmitotic DA neurons to maintain synaptic tiling. (A) Experimental design of the postembryonic degradation of UNC-4 and UNC-37 using the AID system. (B, C) Representative images of synaptic tiling after early postembryonic degradation in *unc-4(miz40); mizSi3* (B) and *unc-37(miz36); mizSi3* (C). (D, E) Representative images of synaptic tiling after late postembryonic degradation in *unc-4(miz40); mizSi3* (D) and *unc-37(miz36); mizSi3* (E). The overlap between the DA8 and DA9 synaptic domains are highlighted with yellow lines. Asterisks: DA9 cell

*Figure 5 continued on next page*

**Figure 5 continued**

body. Scale bar: 10 µm. (F) Quantification of overlap of DA8 and DA9 synaptic domains after postembryonic auxin treatment. See **Figure 5—source data 1**. Each dot represents a single animal. Black bars indicate mean ± SEM. n.s.: not significant; *:p<0.05; ****:p<0.0001.

The online version of this article includes the following source data for figure 5:

**Source data 1.** Quantification of overlap of DA8 and DA9 synaptic domains after postembryonic auxin treatment.

---

also highlights that the AID system is an effective method to uncover the novel spatiotemporal functions of well-characterized genes. Indeed, recent work using the AID system showed that the BRN3-type POU homeobox gene *unc-86/Brn3a* is required for both the initiation and maintenance of multiple neuronal identities in *C. elegans* (**Serrano-Saiz et al., 2018**). The combination of CRISPR/Cas9 genome editing technologies and the AID system will allow us to examine the spatiotemporal functions of any genes, including the genes whose null mutants are lethal.

Neuronal cell fate transformations often lead to altered synaptic connectivity. The temporal transcription factor Hunchback in *Drosophila* controls the neuroblast lineage NB7-1 that gives rise to the

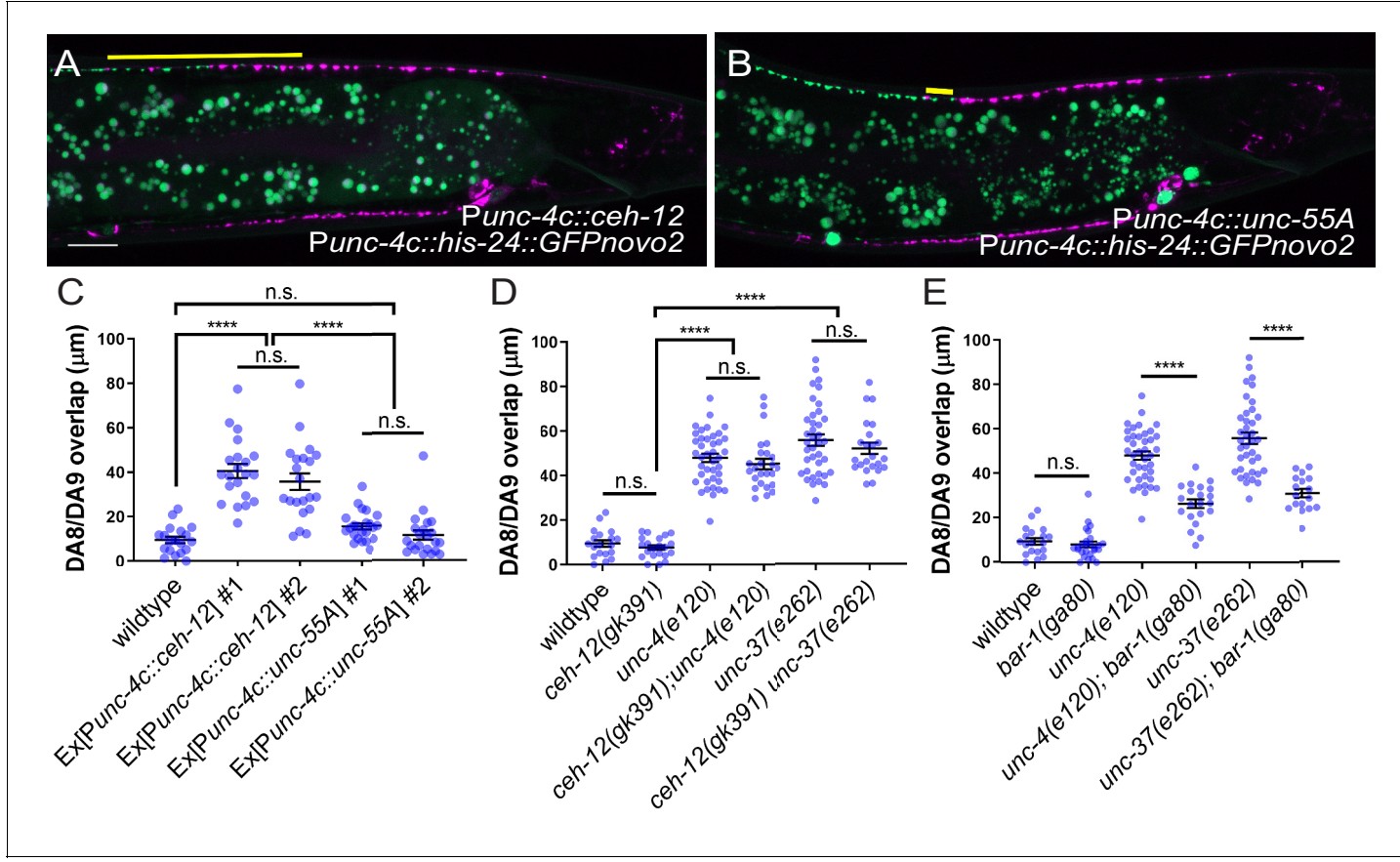

**Figure 6.** *unc-4* and *unc-37* inhibit canonical Wnt signaling to regulate synaptic tiling. (A, B) Representative image of synaptic tiling in ectopic expression of *ceh-12* under the *unc-4c* promoter (A) and *unc-55A* under the *unc-4c* promoter (B). The overlap between the DA8 and DA9 synaptic domains are highlighted with yellow lines. Scale bar: 10 µm. (C) Quantification of overlap between DA8 and DA9 synaptic domains. Two independent transgenic lines were quantified for each array. See **Figure 6—source data 1**. (D) Quantification of overlap between DA8 and DA9 synaptic domains in *ceh-12* mutant animals. See **Figure 6—source data 2**. (E) Quantification of overlap between DA8 and DA9 synaptic domains in *bar-1* mutant animals. See **Figure 6—source data 3**. Each dot represents a single animal. Black bars indicate mean ± SEM. n.s.: not significant; ****:p<0.0001.

The online version of this article includes the following source data and figure supplement(s) for figure 6:

**Source data 1.** Quantification of overlap between DA8 and DA9 synaptic domains.
**Source data 2.** Quantification of overlap between DA8 and DA9 synaptic domains in *ceh-12* mutant animals.
**Source data 3.** Quantification of overlap between DA8 and DA9 synaptic domains in *bar-1* mutant animals.
**Figure supplement 1.** *ceh-12* expression is derepressed in DA8/9 of *unc-4(e120)* and *unc-37(e262)* mutants.

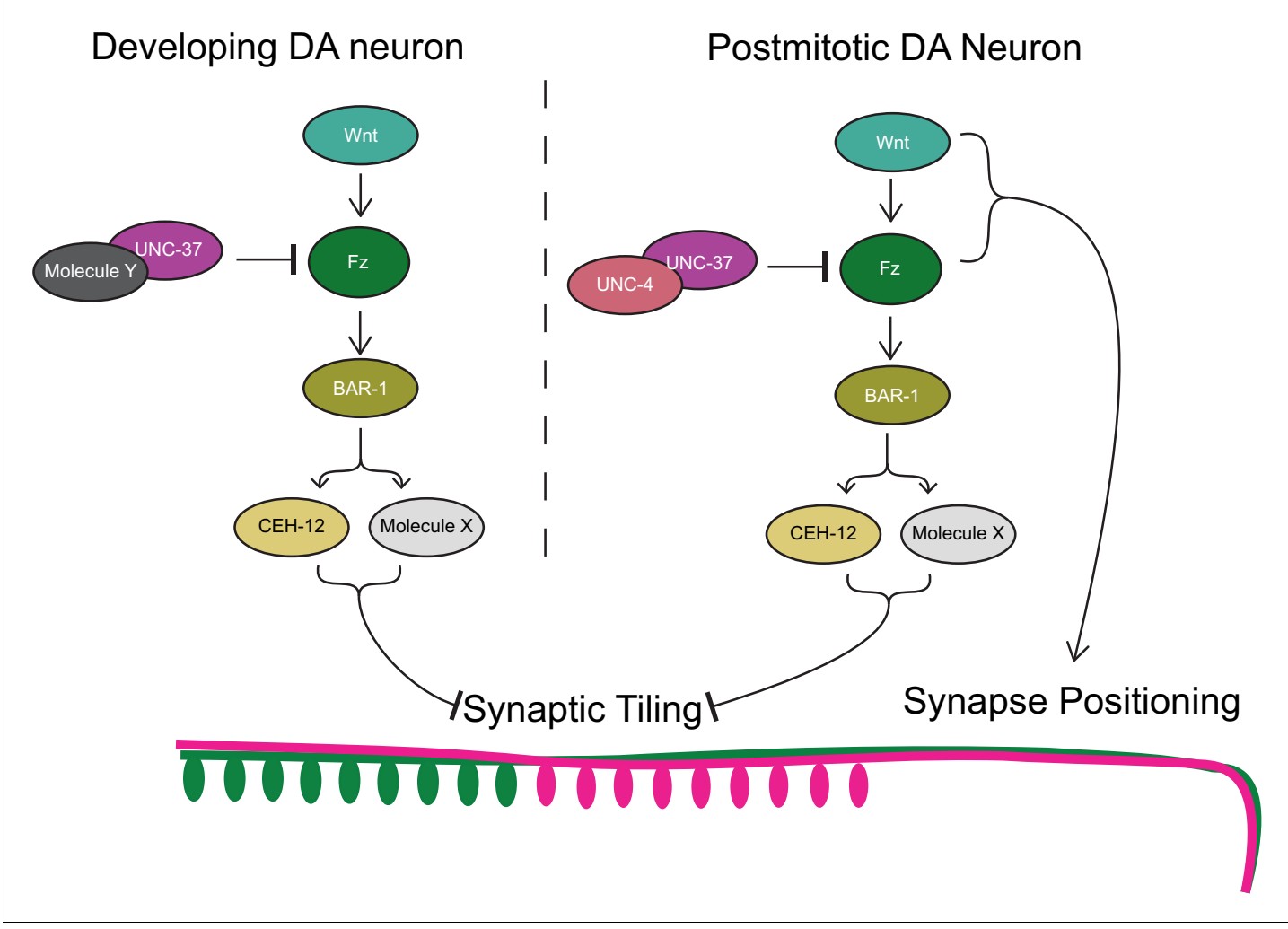

**Figure 7.** Model of UNC-4 and UNC-37 functions in synaptic tiling. UNC-4 functions with UNC-37 in postmitotic DA neurons to regulate synaptic tiling by inhibiting canonical Wnt signaling. Ectopic expression of CEH-12 and other unknown molecules in DA neurons results in synaptic tiling defects. UNC-37 likely forms a repressor complex with an unknown transcription factor in developing DA neurons by inhibiting canonical Wnt signaling to regulate tiled synaptic innervation. Wnt and Fz also function in synapse positioning (*Klassen and Shen, 2007*; *Mizumoto and Shen, 2013b*).

U1-5 motor neurons which innervate the dorsal (U1-2) and ventral (U3-5) muscles (*Isshiki et al., 2001*; *Kohwi and Doe, 2013*). Perturbed expression of Hunchback in the NB7-1 lineage results in the serial formation of U1 neurons, all of which project to the dorsal muscles (*Isshiki et al., 2001*; *Pearson and Doe, 2003*; *Seroka and Doe, 2019*). Likewise, *unc-4* and *unc-37* mutants exhibit synaptic specificity defects in the VA motor neurons due to the partial cell fate transformation of the VA neurons to VB neurons (*Miller et al., 1992*; *White et al., 1992*; *Winnier et al., 1999*). Similarly, *Drosophila* Unc-4 is required in the 7B neurons for the projections into the ipsilateral and contralateral leg neuropils (*Lacin et al., 2020*). This miswiring phenotype is also likely due to the cell fate transformation because Unc-4 is specifically required during development (*Lacin et al., 2020*). Our study showed that the presynaptic patterning defects of DA8 and DA9 in *unc-4* and *unc-37* was not accompanied with major cell fate defects. Our data therefore suggest that the synaptic patterning defects in the mutants of cell fate determinants could occur independent of the cell fate transformation.

The lack of apparent cell fate specification defects in the DA8 and DA9 neurons of *unc-4* and *unc-37* contradicts the previous works showing their critical functions in the specification of the A-type motor neurons (*Kerk et al., 2017*; *Winnier et al., 1999*). One possible explanation is that the mechanisms of the DA8 and DA9 cell fate specification is slightly distinct from the anterior DA neurons.

Consistently, these two most posterior DA neurons, especially DA9, have distinct gene expression profiles compared with the other DA neurons, as exemplified by the series of DA9-specific markers (*Kratsios et al., 2017*).

Similar to DA neurons, VA neurons also exert tiled synaptic innervation (*White et al., 1976*). Because *unc-4* and *unc-37* have been shown to play crucial roles in the AVA>VA wiring specificity primarily during the development of VA neurons (*Miller et al., 1992*; *Winnier et al., 1999*), it would be interesting to examine if *unc-4* and *unc-37* also play a role in VA synaptic tiling. Unfortunately, we could not test this idea as the *mizIs3* marker that should label the synapses of VA11 and VA12, with GFPnovo2::RAB-3 and mCherry::RAB-3, respectively, has variable expression in wild type. Nevertheless, the potential role for *unc-4* in the maintenance of AVA>VA synaptic connection in the L2–L3 stage (*Miller et al., 1992*) is consistent with our discovery of the post-mitotic functions of *unc-4* and *unc-37* in precise neuronal pattern formation. Therefore, it is possible that *unc-4* and *unc-37* also have the additional role of regulating the synaptic tiling in VA neurons as well.

Could the abnormal synaptic tiling pattern of the DA8 and DA9 neurons also be a synaptic specificity defect between the presynaptic DA neurons and the postsynaptic muscle cells? While we do not know which muscles DA8 and DA9 innervate in wild-type and *unc-4* mutants, several observations argue against this. First, our results show that *unc-4* is specifically required in the postmitotic DA neurons after DA cell fates are set. Second, there is little contribution from the post-synaptic muscles for the tiled synaptic patterning of the DA neurons. Previously, we showed that the tiled synaptic pattern of DA8 and DA9 is solely dependent on the physical interaction between the axons of DA8 and DA9 which is mediated by Semaphorin–Plexin signaling (*Mizumoto and Shen, 2013a*). We also showed that the positions of DA8 and DA9 synapses are determined by the Wnt gradient cues rather than the position of the post-synaptic muscle cells (*Klassen and Shen, 2007*; *Mizumoto and Shen, 2013b*). Therefore, it is unlikely that the synaptic tiling defects we observed in *unc-4* and *unc-37* mutants are the synaptic specificity defects between the DA neurons and the postsynaptic muscle cells.

Our results showed that *unc-4* is exclusively required in the differentiated postmitotic DA neurons, while *unc-37* functions in both developing and differentiated DA neurons. This raises the possibility that UNC-37 functions with other transcription factors during neurogenesis of the DA neurons to control tiled synaptic innervation (*Figure 5*). The Groucho family of transcriptional corepressors are known to function with various transcription factors during animal development (*Gasperowicz and Otto, 2005*). In addition to UNC-4, UNC-37 interacts with eh1 domain-containing homeobox proteins (COG-1 and MAB-9) (*Chang et al., 2003*; *Jafari et al., 2011*; *Winnier et al., 1999*) and POP-1/T-cell factor (*Calvo et al., 2001*) to regulate cell fate specifications during development. UNC-37 may function with these transcription factors and/or unidentified UNC-37-interacting transcription factors during development of the DA neurons. Further genetic and biochemical studies to identify UNC-37 binding partners would help better understand the functions of *unc-37* in synapse patterning.

UNC-4 inhibits canonical Wnt signaling to repress *ceh-12* expression in VA neurons (*Schneider et al., 2012*; *Von Stetina et al., 2007*). Consistently, loss of *egl-20/wnt* or *mig-1/frizzled* in *unc-4* mutants suppresses the ectopic expression of *ceh-12* in the VA neurons and restores the VA-AVA connections (*Schneider et al., 2012*). We also showed that loss of *bar-1/β-catenin* partially restores the synaptic tiling defects of *unc-4* and *unc-37* mutants. On the other hand, unlike previous work that showed that *ceh-12* mutants can suppress the synaptic connectivity defects between VA and AVA neurons in *unc-4* mutants, we did not observe significant suppression of the synaptic tiling defects in the *ceh-12(gk391); unc-4(e120)* and *ceh-12(gk391) unc-37(e262)* double mutants. This suggests that *ceh-12* is not the only Wnt effector gene that regulates synaptic tiling and is upregulated in the *unc-4* and *unc-37* mutants. Future single-cell RNA sequencing of the DA8 and DA9 neurons in *unc-4* mutants, as well as genetic suppressor screening of the synaptic tiling defects in the *ceh-12; unc-4* and *ceh-12 unc-37* mutants will be necessary to identify additional factors that function redundantly with *ceh-12*.

The genetic interaction between *bar-1* and *unc-4* (and *unc*-37) indicates that the canonical Wnt signaling pathway is required for the proper tiled synaptic innervation in the DA neurons. Specifically, Wnt signaling functions as a negative regulator of synaptic tiling (*Figure 7*) which is controlled by Sema–Plexin signaling. Sema–Plexin signaling has been shown to function as a negative regulator of synapse formation in both *C. elegans* and mice (*Duan et al., 2014*; *Mizumoto and Shen, 2013a*;

*Tran et al., 2009*). It is therefore possible that Wnt signaling functions as a prosynaptogenic cue by inhibiting Sema/Plexin signaling. Indeed, Wnt-7a has been shown to induce synaptogenesis (*Hall et al., 2000*; *Lucas and Salinas, 1997*). Uncovering the Wnt effector genes that control synaptic tiling would therefore help us understand the molecular functions of the canonical Wnt signaling in synapse formation.

In addition to the function of canonical Wnt signaling in synaptic tiling, we and others have shown that two Wnts, LIN-44 and EGL-20, function as gradient inhibitory cues for the proper positioning of DA8 and DA9 synapses (*Klassen and Shen, 2007*; *Mizumoto and Shen, 2013b*). Disruption of the Wnt gradient patterns affect the severity of the synaptic tiling defects of *plx-1* mutants, due to the displaced synapse position of DA8 and DA9 (*Mizumoto and Shen, 2013b*). For example, *egl-20* enhances the synaptic tiling defects of *plx-1* mutants due to the posterior displacement of the DA8 synaptic domain. The function of Wnt gradient signal in synapse positioning is mediated by previously uncharacterized non-canonical pathways as it does not require canonical Wnt pathway components including *bar-1/β-catenin* (*Klassen and Shen, 2007*). Our results therefore suggest that canonical Wnt signaling and non-canonical Wnt signaling cooperatively regulate the precise synapse patterning in *C. elegans*.

## Materials and methods

**Key resources table**

| Reagent type (species) or resource | Designation | Source or reference | Identifiers | Additional information |
|---|---|---|---|---|
| Gene (*Caenorhabditis elegans*) | *unc-4* | WormBase | F26C11.2 | |
| Gene (*Caenorhabditis elegans*) | *unc-37* | WormBase | W02D3.9 | |
| Strain, strain background (*C. elegans*) | *unc-4(e120)* | *C. elegans* stock center (CGC) | CB120 | |
| Strain, strain background (*C. elegans*) | *unc-4(e2322)* | *C. elegans* stock center (CGC) | NC37 | |
| Strain, strain background (*C. elegans*) | *unc-37(e262)* | *C. elegans* stock center (CGC) | CB262 | |
| Strain, strain background (*C. elegans*) | *ceh-12(gk391)* | *C. elegans* stock center (CGC) | VC995 | |
| Strain, strain background (*C. elegans*) | *bar-1(ga80)* | *C. elegans* stock center (CGC) | EW15 | |
| Strain, strain background (*C. elegans*) | *unc-4(miz40)* | This study | UJ1017 | *unc-4::AID::BFP* |
| Strain, strain background (*C. elegans*) | *unc-37(miz36)* | This study | UJ1013 | *unc-37::AID::BFP* |
| Strain, strain background (*C. elegans*) | *mizIs3* | This study | UJ124 | DA8/DA9 synaptic marker |
| Strain, strain background (*C. elegans*) | *mizSi3* | This study | UJ1133 | P*unc-4c::TIR1* |

### Strains

Bristol N2 strain was used as a wild-type reference. All strains were cultured in the nematode growth medium (NGM) with OP50 as described previously (*Brenner, 1974*). Unless noted, all strains were maintained at room temperature (22°C). The following alleles were used in this study: *unc-4(e120)*, *unc-4(e26)*, *unc-4(e2322ts)*, *unc-4(miz40)*, *unc-37(e262)*, *unc-37(miz36)*, *plx-1(nc36)*, *ceh-12(gk391)*, *bar-1(ga80)*. Genotyping primers are listed in the supplemental material.

### Transgenes

The following transgenes were used in this study: *mizIs3 (Punc-4::zf1-GFPnovo2::rab-3, Pmig-13::zif-1, Pmig-13::mCherry::rab-3, Podr-1::RFP); wyIs320 (Pitr-1::plx-1::GFP, Pmig-13::mCherry::rab-3,*

*Podr-1::GFP); mizEx362 (Pplx-1::GFP, Punc-4c::his-24::mCherry, Podr-1::GFP); mizEx365 (Prap-2:: GFP, Punc-4c::his-24::mCherry, Podr-1::GFP); mizEx410 (Punc-4c::his-24::mCherry, Podr-1::GFP); hdIs1 (Punc-53::GFP, rol-6)* (**Kerk et al., 2017**); *mizEx396 (Punc-129DB::his-24::mCherry, Punc-4c:: his-24::GFPnovo2, Podr-1::GFP); wdEx419(Pacr-16::GFP, rol-6(su1006))* (**Winnier et al., 1999**); *otIs476 (Pglr-4::TagRFP)* (**Kratsios et al., 2017**); *wdIs62 (Pceh-12::GFP, unc-119(+))* (**Von Stetina et al., 2007**); *mizEx394, mizEx395 (Punc-4c::ceh-12, Punc-4c::his-24::GFPnovo2, Podr-1::GFP); mizEx408, mizEx409 (Punc-4c::unc-55A, Punc-4c::his-24::GFPnovo2, Podr-1::GFP); mizEx434, mizEx435 (Pplx-1::plx-1::GFP, Punc-4c::his-24::mCherry, Podr-1::GFP); mizEx436 (Punc-4c::his-24:: GFPnovo2, Podr-1::GFP); mizEx437, mizEx439 (Punc-4c::unc-4, Punc-4c::his-24::GFPnovo2, Podr-1:: GFP); mizEx440, mizEx441 (Punc-4c::unc-37, Punc-4c::his-24::GFPnovo2, Podr-1::GFP).*

The transgenic lines with extrachromosomal arrays were generated using the standard microinjection method (**Fire, 1986**; **Mello et al., 1991**). The integration of the extrachromosomal arrays into the chromosomes was conducted by standard UV irradiation method (**Evans, 2006**) For a DA8/DA9 synaptic tiling marker (*mizIs3*), we used the ZIF-1/ZF1 degradation system (**Armenti et al., 2014**). The synaptic vesicle-associated protein, RAB-3, was fused with GFPnovo2, a codon-optimized brighter variant of GFP (**Hendi and Mizumoto, 2018**), and ZF1 degron from PIE-1, and expressed under the *unc-4* promoter. mCherry::RAB-3 and ZIF-1 are expressed under the DA9-specific *mig-13* promoter. ZIF-1 expressed in DA9 leads the ZF1::GFPnovo2::RAB-3 fusion protein to the ubiquitin-mediated protein degradation thereby creating a better color separation of the DA8 synapses labeled with GFPnovo2 and DA9 synapses labeled with mCherry.

## Plasmid construction

*C. elegans* expression clones were made in a derivative of pPD49.26 (A. Fire), the pSM vector (a kind gift from S. McCarroll and C. I. Bargmann). *unc-4, unc-37, ceh-12,* and *unc-55* cDNAs were amplified with Phusion DNA polymerase (NEB) from N2 cDNA library synthesized with Superscript III first-strand synthesis system (Thermo Fisher Scientific). The amplified cDNAs were cloned into the AscI and KpnI sites of pSM vector using Gibson assembly method (**Gibson, 2011**).

The *loxP_myo-2_NeoR* dual-selection cassette vector (**Au et al., 2019**; **Norris et al., 2015**) was used to construct the repair template plasmids for *unc-4::AID::BFP* and *unc-37::AID::BFP* strains. AID and codon-optimized BFP sequences were cloned into the *SacII* site of the *loxP_myo-2_NeoR* plasmid using Gibson assembly (**Gibson, 2011**). The 5' and 3' homology arms of *unc-4* and *unc-37* were amplified from N2 genomic DNA using Phusion DNA polymerase and cloned into the *SacII* and *NotI* sites, respectively, by Gibson assembly.

The *loxP_myo-2_NeoR* was also used to construct the repair template plasmid for single copy insertion of P*unc-4c::TIR1* at oxTi365 MosSCI site. Previously reported 5' and 3' homology arms of oxTi365 MosSCI site (**Obinata et al., 2018**) were amplified from N2 genomic DNA using Phusion DNA polymerase and cloned into the *SacII* and *NotI* sites, respectively, by Gibson assembly. The *SrfI* recognition sequence (GCCCGGGC) was inserted at the 3' end of the 5' homology arm. TIR1 from pLZ31 (Addgene #71720) was amplified and cloned into *AscI* and *KpnI* sites of the pSM plasmid containing the *unc-4c* promoter at the *SphI* and *AscI* sites. P*unc-4c::TIR1::unc-54 utr* from this plasmid was amplified and cloned into the *SrfI* site of the repair template by Gibson assembly.

pTK73 plasmid (**Obinata et al., 2018**) was used as a backbone vector for gRNA expression plasmids construction. The following gRNAs were used: *unc-4#1:* TTTGAACCGTGCCCTCCAT, *unc-4#2:* GCGACTAATGCATTGACTA, *unc-37#1:* ACTGGATCAGGGGAGAAGA, *unc-37#2:* CGATTATCG TACAGAATAG, and oxTi365: CGGTCGCGGTACTCCAAAT.

CRISPR/Cas9 genome editing *unc-4(miz40[unc-4::AID::BFP]), unc-37(miz36[unc-37::AID::BFP]),* and *mizSi3 (Punc-4c::TIR1-loxP-Pmyo-2::GFP-NeoR-loxP)* were generated using a dual-selection method as previously described (**Au et al., 2019**; **Norris et al., 2015**). To generate *unc-4(miz40), unc-4* repair template plasmid, two *unc-4* sgRNA plasmids and Cas9 plasmid (Addgene #46168) (**Friedland et al., 2013**) were co-injected into young adults. To generate *unc-37(miz36), unc-37* repair template plasmid, two *unc-37* sgRNA plasmids and Cas9 plasmid were co-injected into young adults. The screening of the genome edited animals was conducted according to previous works (**Au et al., 2019**; **Norris et al., 2015**). Briefly, F1 progenies from the injected animals were treated with Geneticin (G418) (Sigma-Aldrich) (50 mg/mL) for NeoR selection (**Au et al., 2019**). The fully penetrant, uniform, and low expression of P*myo-2::GFP* in the pharynx was used to select the genome-edited candidates. The selection cassette was excised by injecting Cre recombinase

plasmid (pDD104, Addgene #47551) and screened for animals that lack P*myo-2::GFP* expression. The successful genome editing was confirmed by Sanger sequencing.

For the generation of *mizSi3 (Punc-4c::TIR1-loxP-Pmyo-2::GFP-NeoR-loxP)*, we injected P*unc-4c:: TIR1* repair plasmid, *oxTi365* gRNA, and Cas9 plasmid into wild-type animals and screened animals with uniform and penetrant P*myo-2::GFP* expression. For simplicity, we label *mizSi3(Punc-4c::TIR1-loxP-Pmyo-2::GFP-NeoR-loxP)* as *mizSi3(Punc-4c::TIR1)*.

### Neuronal cell fate marker expression

The following transgenes were used as cell fate markers: *hdIs1*[P*unc-53::GFP*] (DA-specific), *mizEx396*[P*unc-129DB::his-24::mCherry*, P*unc-4c::his-24::GFPnovo2*] (DB-specific), *wdEx419*[P*acr-16::GFP*] (DB-specific), *wdEx60*[P*acr-5::GFP*] (DB-specific), *wyIs320*[P*itr-1::plx-1::GFP*, P*mig-13:: mCherry::rab-3*] (DA9-specific), *otIs476*[P*glr-4::TagRFP*] (DA9-specific), *mizIs3*[P*unc-4c::zf1-GFPnovo2::rab-3*, P*mig-13::zif-1*, P*mig-13::mCherry::rab-3*] (DA9 specific). mCherry::RAB-3 expression in *mizIs3* was used as an indicator of the *mig-13* promoter activity. PLX-1::GFP signal was used to examine the *itr-1* promoter activity. Scoring of the DA-, DB-, DA9 cell fate marker expressions in DA8 and DA9 neurons was conducted using ZEISS Axioplan two fluorescent microscope, except P*itr-1* in *unc-4(e120)* mutants, which was examined under the ZEISS LSM800 confocal microscope.

### Temperature shift assay

For embryonic knockdown, the L4 parental animals of *unc-4(e2322ts)* mutants were transferred to 25°C (restrictive temperature). Newly hatched L1 animals from the parental animals were transferred to 16°C (permissive temperature) until L4 to examine the synaptic tiling of DA8 and DA9 neurons. For postembryonic knockdown, the L4 parental animals of *unc-4(e2322ts)* mutants were transferred to 16°C (permissive temperature) at L4 stage. Newly hatched L1 animals were transferred to 25°C (restrictive temperature) until L4 to examine the synaptic tiling of DA8 and DA9 neurons. Animals that were not transferred were used as controls. Wild-type controls were cultured at 16°C and 25°C.

### Spatiotemporal degradation of UNC-4 and UNC-37 using the AID system

The synthetic auxin analog, α-napthaleneacetic acid (K-NAA) (*Martinez et al., 2020*), was dissolved in dH$_2$O to prepare a 400 mM stock solution. The 4 mM working solution of K-NAA was prepared by diluting a stock solution with M9 buffer. 1000 µL of 4 mM K-NAA was added to the NGM plates with OP50 bacteria and were allowed to dry overnight at room temperature. To induce protein degradation, experimental animals were transferred onto the K-NAA plates and were kept at room temperature. As a control, animals were transferred onto the control (M9) plates. For embryonic degradation of UNC-4 and UNC-37, the L4 parental *unc-4(miz40);mizSi3* and *unc-37(miz36);mizSi3* animals were transferred to K-NAA plates at L4 stage. Newly hatched L1 animals from the parental animals were transferred to the control (M9) plates until L2 or L4 to examine the synaptic tiling of DA8 and DA9 neurons. For postembryonic degradation of UNC-4 and UNC-37, the L4 parental *unc-4(miz40);mizSi3* and *unc-37(miz36);mizSi3* animals were transferred to the control (M9) plates at L4 stage. Newly hatched L1 (early postembryonic degradation) or mid-late L2 (late postembryonic degradation) animals from the parental animals were transferred to K-NAA plates until L4 to examine the synaptic tiling of DA8 and DA9 neurons.

### Confocal microscopy

Images of fluorescently tagged fusion proteins were captured in live *C. elegans* using a Zeiss LSM800 Airyscan confocal microscope (Carl Zeiss, Germany) with oil immersion lens 63× magnification (Carl Zeiss, Germany). Worms were immobilized on 2.5% agarose pad using a mixture of 7.5 mM levamisole (Sigma-Aldrich) and 0.225 M BDM (2,3-butanedione monoxime) (Sigma-Aldrich). Images were analyzed with Zen software (Carl Zeiss) and ImageJ (NIH, USA). Twenty to 26 Z-stack images were taken for each animal to encompass the cell bodies, axons, and synapses of the DA8 and DA9 neurons. The definition of DA8/DA9 synaptic overlap was defined by the distance between the most anterior DA9 synapse and the most posterior DA8 synapses (*Mizumoto and Shen, 2013a*; *Chen et al., 2018*). L4.4–L4.5 larval stage animals, judged by the stereotyped shape of the developing vulva (*Mok et al., 2015*), were used for quantification.

## Statistics

Prism9 (GraphPad Software, USA) was used for statistical analyses. We applied the one-way ANOVA method with post hoc Tukey's multiple comparison test for comparison among three or more parallel groups with multiple plotting points, and chi-square test (with Yates' continuity corrected) for comparison between two binary data groups. Data were plotted with error bars representing standard errors of mean (SEM). *, **, and *** represent p-value < 0.05, < 0.01, and < 0.001, respectively.

## Acknowledgements

We are grateful to David Miller and Don Moerman for suggestions on the project. We also thank Kenji Sugioka, Riley St. Clair, and Ardalan Hendi for comments on the manuscript. Some strains used in this study were obtained from the *Caenorhabditis* Genetics Center (CGC), which is funded by NIH Office of Research Infrastructure Programs (P40 OD010440) and *C. elegans* gene knockout consortium. This project is funded by HFSP (CDA-00004/2014) and CIHR (PJT-148667). KM is a recipient of Canada Research Chair and Michael Smith Foundation for Health Research Scholar. JW is a recipient of the NSERC Undergraduate Student Research Awards.

## Additional information

### Funding

| Funder | Grant reference number | Author |
| --- | --- | --- |
| Human Frontier Science Program | CDA-00004/2014 | Kota Mizumoto |
| Canadian Institutes of Health Research | PJT-148667 | Kota Mizumoto |
| Canada Research Chairs | 950-232435 | Kota Mizumoto |
| Michael Smith Foundation for Health Research | SCH-2017-2001 | Kota Mizumoto |
| NSERC | Undergraduate Student Research Awards | Jane Wang |

The funders had no role in study design, data collection and interpretation, or the decision to submit the work for publication.

### Author contributions

Mizuki Kurashina, Conceptualization, Formal analysis, Investigation, Visualization, Methodology, Writing - original draft, Writing - review and editing; Jane Wang, Formal analysis, Validation, Investigation, Methodology; Jeffrey Lin, Formal analysis, Writing - review and editing; Kathy Kyungeun Lee, Resources, Investigation, Methodology; Arpun Johal, Investigation, Methodology; Kota Mizumoto, Conceptualization, Formal analysis, Supervision, Funding acquisition, Validation, Investigation, Methodology, Writing - original draft, Writing - review and editing

### Author ORCIDs

Kota Mizumoto  https://orcid.org/0000-0001-8091-4483

### Decision letter and Author response

Decision letter https://doi.org/10.7554/eLife.66011.sa1
Author response https://doi.org/10.7554/eLife.66011.sa2

## Additional files

### Supplementary files

• Transparent reporting form

Data availability

All data generated or analysed during this study are included in the manuscript and supporting files.

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

## Appendix 1

## Supplemental information

Genotyping primers:

> plx-1(nc36)
> Forward: CTTCGAGAGCCCCCTCATTCTTAATG
> Reverse: CCGGCACACGTTAAACTAGTGCTACCG
> ceh-12(gk391)
> Forward: TTGCCGCACATCTCTCATAC
> Mutant Reverse: GCTTCTCGTCTTCTCGTCCT
> Wild-type Reverse: GATGAGAAGACCACGAACCG
> *bar-1(ga80)* (mutant pcr product can be digested with *Mse*I)
> Forward: GTGAGTTCTGGAATTGCTCG
> Reverse: GATGACGGATGTAGAGATGA
> unc-4(miz40)
> Forward: AGTCTCCCCCTCTTCCAACC
> Mutant Reverse: ACATGGAAGGAACCGTGGAC
> Wild-type Reverse: CGTTTCATCTCTCTCATCTC
> unc-37(miz36)
> Forward: GCTCCGTTCTCTCATGTGAC
> Mutant Reverse: ACATGGAAGGAACCGTGGAC
> Wild-type Reverse: CGTTTCATCTCTCTCATCTC

Primers for plasmid construction:

> *unc-4* cDNA (cloned into *Ascl/Kpn*I sites of the ΔpSM vector)
> Forward: CTTCTTCTTCCAAAAAGTGAAAGGCGCGCCATGATCGGTGCACTGCATGC
> Reverse: GGAGCTCAGATATCAATACCATGGTACCTTATACACTTTTCAGTAATTCAGC
> *unc-37* cDNA (cloned into *Ascl/Kpn*I sites of the ΔpSM vector)
> Forward: CTTCTTCTTCCAAAAAGTGAAAGGCGCGCCATGAAGGCATCGTATCTGGAAACC
> Reverse: GCGGAGCTCAGATATCAATACCATGGTACCTTAATATTCAACTGCATAGAG
> *ceh-12* cDNA (cloned into *Ascl/Kpn*I sites of the ΔpSM vector)
> Forward: CTTCTTCTTCCAAAAAGTGAAAGGCGCGCCCATGATGTTTTCCTCAATAGA
> Reverse: CAATCAACTTCCTCTTCTTGAGGTACCATGGTATTGATATCTGAGCTC
> *unc-55* cDNA (cloned into *Ascl/Kpn*I sites of the ΔpSM vector)
> Forward: CTTCTTCTTCCAAAAAGTGAAAGGCGCGCCATGCAGGATGGCTCATCAGG
> Reverse: CAACTACCTTCCAGAAATTAGGGTACCATGGTATTGATATCTGAGCTC

CRISPR:

> *unc-4* gRNA
> Forward 1: TCTTGTTTGAACCGTGCCCTCCAT
> Reverse 1: AAACATGGAGGGCACGGTTCAAAC
> Forward 2: TCTTGGCGACTAATGCATTGACTA
> Reverse 2: AAACTAGTCAATGCATTAGTCGCC
> *unc-4* repair template
> 5' Forward: TAAAACGACGGCCAGTGAATTCCCGCGGAGGATGCTATGGTGGACCAG
> 5' Reverse: TTAGGGGCTCCGGCTCCGGCTCCTACACTTTTCAGTAATTCAGCAA
> 3' Forward: AGTTATAGTTGCAGGACCACTATTTTTTTAAAATTCAATTTTGAACCG
> 3' Reverse: CATGATTACGCCAAGCTTGCGGCCGCGCTCTGCGAGACGTCACTGTCATC
> *unc-37* gRNA
> Forward 1: TCTTGACTGGATCAGGGGAGAAGA
> Reverse 1: AAACTCTTCTCCCCTGATCCAGTC
> Forward 2: TCTTGCGATTATCGTACAGAATAG
> Reverse 2: AAACCTATTCTGTACGATAATCGC
> *unc-37* repair template
> 5' Forward: AACGACGGCCAGTGAATTCCCGCGGCTTGCACTCCCTATTAGTATTGC
> 5' Reverse: GGGGCTCCGGCTCCGGCTCCATATTCAACTGCATAGAGAGTTG
> 3' Forward: AGTTATAGTTGCAGGACCACTTTTTTTTCCGCAGTGAATTTATGATC
> 3' Reverse: ACCATGATTACGCCAAGCTTGCGGCCGCAATGGGTGGAGCCTTGGAGATA
> *AID::BFP* cassette

Forward: GACGGCCAGTGAATTCCCGCGGGGAGCCGGAGCCGGAGCCGGAGCCCCTA
Reverse: ATAGGCCGCCTGATGCGCTAATTAAGCTTGTGACCCAGTTTG
oxTi365 gRNA
Forward: TCTTGCGGTCGCGGTACTCCAAAT
Reverse: AAACATTTGGAGTACCGCGACCGC
oxTi365 repair template
5' Forward: AAACGACGGCCAGTGAATTCCCGCGGAGCGTTGATGATTGGAGGGG
5' Reverse: AAGTTATAGGCCGCCTGATGCCCGGGCATTTGGAGTACCGCGACCGT
3' Forward: GTTATAGTTGCAGGACCACTAGGCAGGATATTCTCCATTTCTG
3' Reverse: CATGATTACGCCAAGCTTGCGGCCGCGGGAATGTGCGTGGTGTTTG
P*unc-4c::TIR1* cassette (inserted into Srf1 site via Gibson assembly)
Forward: CGCGGTACTCCAAATGCCCATGACCATGATTACGCCAAG
Reverse: TTATAGGCCGCCTGATGCCCAAACGCGCGAGACGAAAGGG

