## [Decision Letter]

**Acceptance summary:**

The reviewing team recognizes the importance of defining postmitotic function of transcription factor. They also found it interesting that your data showed that unc-4 and unc-37 function differently in synapse tiling. Overall, this study provided new insights in the molecular mechanisms of synapse tiling.

**Decision letter after peer review:**

Thank you for submitting your article "Sustained expression of unc-4 homeobox gene and unc-37/Groucho specifies the spatial synaptic organization in *C. elegans*" for consideration by *eLife*. Your article has been reviewed by 2 peer reviewers, and the evaluation has been overseen by Kang Shen as the Reviewing Editor and K VijayRaghavan as the Senior Editor. The reviewers have opted to remain anonymous.

Both reviewers found your manuscript interesting and potentially suitable for publication in e*Life*. One reviewer summarized your findings as below. This manuscript by Kurashina et al. describes a novel post-mitotic role in synaptic patterning for a cell fate determining gene, unc-4, and its co-repressor, unc-37. The DA neurons in *C. elegans* are cholinergic motor-neurons that exhibit unique synaptic tiling of their dorsal axonal segments. Mizumoto et al. has previously shown that Semaphorin-Plexin signaling is required to establish the tiling between DA8 and DA9, by functioning in cis in the DA9 neuron. Using temperature-sensitive mutant of unc-4, as well as a combination of CRISPR/Cas9 genome editing with the AID system for specific temporal degradation, the authors nicely examine the spatiotemporal requirement of unc-4, and show that unc-4 is required only post-mitotically for synapse tiling, but not during the development of the DA neurons. Interestingly, activity of the corepressor unc-37 is required both during development and postmitotically for correct tiling. unc-4 and unc-37 are suggested to function by inhibiting the canonical wnt signaling. Overall this is an interesting study that sheds light on our understanding of the post-developmental role of cell fate genes in synapse patterning.

The reviewers did suggest the following essential revisions.

Essential revisions:

1) The authors present in their introduction the results from Kerk et al., regarding the role of unc-4 as a cell fate determining gene for the VAs and DAs. Kerk et al. have shown that UNC-4 is specifically required for the expression of DA genes, without affecting ACh pathway genes. Table 1, however, doesn't fully recapitulate the same results and actually shows that unc-4 and unc-37 mutants do not exhibit significant cell fate defects. The authors use these results to argue in the discussion that the synaptic patterning defects can occur independent of the cell fate transformation. The issue of unc-4 as a cell fate-determining gene of A type motor neurons needs to be more clearly addressed. The authors should test whether acr-5 expression is elevated in DAs in unc-4 and unc-37 mutants (Winnier 1999, Kerk 2017).

2) Both reviewers had some concerns about the tiling markers. The authors use a clever strategy to assess tiling of individual cholinergic motor neurons using DA8 and DA9 as a model, but in some cases observe variable degradation of the RAB-3::GFPnovo, presumably due to weak expression of ZIF1 in some of the mutants. This makes it a little difficult to assess the tiling defects in some of the figures. The residual GFPnovo signal seems to be defined based on colocalization with the more broadly expressed mCherry::RAB-3 marker, but no data is shown for the extent of colocalization in the absence of ZIF1. This analysis would benefit from more explanation.

In Figure 1, the anterior extension of the DA9 synapses in the unc-4 mutants is quite dramatic. However, all the anterior ectopic puncta shown in Figure 1C seem to co-localize with a green puncta. The authors should make sure that there is no ectopic expression of mig-13 in the DA8 neuron on unc-4 mutant. I don't have a similar concern for the unc-37 mutant because there is little perfect colocalization of the anterior synapses in the unc-37 mutants.

3) Two reviewers suggested to use plx-1::GFP to provide more insight in the phenotype of unc-4. Since the itr-1::plx-1::gfp is too dim in the unc-4 mutants, plx-1::plx-1::gfp (either transgene or knock in) might provide useful insight in how unc-4 controls plx-1. Of course, it could also be that plx-1 is expressed in many cells, in that case, it would be hard to have subcellular resolution.

4) In Figure 2+3, the authors drew the conclusion that unc-4 is not required during embryogenesis for synaptic connection. Since they imaged the animals during L4, it is possible that the unc-4 is required for both early and late for synaptic tiling. They should score the phenotype in late L1 or L2 to make sure that the phenotype is not there in the embryonic KD experiments.

*Reviewer #3 (Recommendations for the authors):*

Do the RAB-3::GFPnovo and RAB-3::mCherry markers always have complete overlap, for example in DA9? I'm wondering if variability in the extent of labeling across the 2 markers could influence the ability to assess tiling defects, for example in cases where the RAB-3::GFP marker is only partially degraded.

The previously defined roles for unc-4/unc-37 focused around AVA connectivity with post-embryonic born VA neurons. The authors provide evidence that a subset of neuronal identity markers in DA8 and DA9 neurons are largely unchanged in unc-4 and unc-37 mutants. Do the authors envision that unc-4 and unc-37 have distinct roles in embryonic born DA neurons from those in post embryonic born VA neurons? Or perhaps additional roles?

The downstream pathways (ceh-12, wnt) identified as important for tiling strongly overlap with those previously identified as important for neuronal fate determination. While perhaps not absolutely critical, identification of the proposed additional Wnt-regulated TF that is important (with ceh-12) for tiling might distinguish specific mechanisms for regulation of tiling and be a strong addition the story.

Related, has tiling been examined in VA neurons? Might unc-4/unc-37 have roles in both neuronal fate determination and later roles in synaptic tiling in these neurons? I don't expect the authors to resolve these questions experimentally, but a more direct discussion might help to clarify.

---

## [Author Response]

Essential revisions:1) The authors present in their introduction the results from Kerk et al., regarding the role of unc-4 as a cell fate determining gene for the VAs and DAs. Kerk et al. have shown that UNC-4 is specifically required for the expression of DA genes, without affecting ACh pathway genes. Table 1, however, doesn't fully recapitulate the same results and actually shows that unc-4 and unc-37 mutants do not exhibit significant cell fate defects. The authors use these results to argue in the discussion that the synaptic patterning defects can occur independent of the cell fate transformation. The issue of unc-4 as a cell fate-determining gene of A type motor neurons needs to be more clearly addressed. The authors should test whether acr-5 expression is elevated in DAs in unc-4 and unc-37 mutants (Winnier 1999, Kerk 2017).

We examined the expression of *acr-5::GFP* using the transgene used by Winnier et al., 1999 and confirmed that *acr-5* is mis-expressed in many of the A-class motor neurons in the ventral nerve cord (see Author response image 1). However, we only found a minor degree of *acr-5* misexpression in the DA8/9 neurons (yellow arrows in Author response image 1) similar to *acr-16.* While we do not have an explanation for why DA8/9 exhibit minor cell fate defects compared with more anterior DA neurons, this result is consistent with our conclusion that the minor cell fate defects of DA8 and DA9 in *unc-4* and *unc-37* mutants are unlikely to account for the severe and fully penetrant synaptic tiling defects. We have included the quantification of *acr-5* expression in Table 1.

2) Both reviewers had some concerns about the tiling markers. The authors use a clever strategy to assess tiling of individual cholinergic motor neurons using DA8 and DA9 as a model, but in some cases observe variable degradation of the RAB-3::GFPnovo, presumably due to weak expression of ZIF1 in some of the mutants. This makes it a little difficult to assess the tiling defects in some of the figures. The residual GFPnovo signal seems to be defined based on colocalization with the more broadly expressed mCherry::RAB-3 marker, but no data is shown for the extent of colocalization in the absence of ZIF1. This analysis would benefit from more explanation.In Figure 1, the anterior extension of the DA9 synapses in the unc-4 mutants is quite dramatic. However, all the anterior ectopic puncta shown in Figure 1C seem to co-localize with a green puncta. The authors should make sure that there is no ectopic expression of mig-13 in the DA8 neuron on unc-4 mutant. I don't have a similar concern for the unc-37 mutant because there is little perfect colocalization of the anterior synapses in the unc-37 mutants.

To exclude the possibility that the synaptic tiling defects observed in *unc-4* mutants is due to the compromised ZIF-1/ZF1 degron system in the *mizIs3* synaptic tiling marker used in this study, we examined the synaptic tiling defects using another synaptic tiling marker, *wyIs446*, that we used in our previous work (Chen et al., 2018 *eLife*). Because *wyIs446* does not use the ZIF-1/ZF1 degron system, both DA8 and DA9 express GFP::RAB-3, while mCherry::RAB-3 is only expressed in DA9. In both wildtype and *unc-4* mutants, DA9 expressed both GFP::RAB-3 and mCherry::RAB-3, and GFP and mCherry puncta are nicely colocalized. In this condition we still observed drastic synaptic tiling defects in the *unc-4* mutants. Therefore, we believe that the synaptic tiling defects observed in *unc-4; mizIs3* is not due to the compromised ZIF-1/ZF1 degron system. We included their representative images and signal histogram of GFP and mCherry signals in Figure 1—figure supplement 1.

We also examined the expression of mCherry::RAB-3 in the DA8 cell body of *unc-4* mutants. In wildtype, upon increasing the image brightness, mCherry::RAB-3 signal is detectable in the DA9 cell body but not in the DA8 cell body. In *unc-4* mutants, we did not observe mCherry signal in the DA8 cell body, suggesting that DA8 does not express the DA9 marker, mCherry::RAB-3. Therefore, we believe that all of the synaptic puncta labeled with mCherry::RAB-3 are DA9 synapses. This result is included in Figure 1—figure supplement 2.

In conclusion, we did not find any evidence that suggests the synaptic tiling defects observed in *unc-4; mizIs3* is due to the misexpression of the markers used in this strain.

3) Two reviewers suggested to use plx-1::GFP to provide more insight in the phenotype of unc-4. Since the itr-1::plx-1::gfp is too dim in the unc-4 mutants, plx-1::plx-1::gfp (either transgene or knock in) might provide useful insight in how unc-4 controls plx-1. Of course, it could also be that plx-1 is expressed in many cells, in that case, it would be hard to have subcellular resolution.

We examined the expression and localization of the endogenous PLX-1 by inserting GFPnovo2 cDNA at the 3’ end of the endogenous *plx-1* locus using CRISPR/Cas9 genome editing, but we did not observe reliable level of expression from the dorsal axons including DA9. The transgenic overexpression of P*plx-1::plx-1::gfp* highlighted the neuronal PLX-1::GFP in DA9 judged based on its commissural localization, but the signals from the other neurons and non-neuronal cells (hypodermis and muscles) masked the subcellular localization of PLX-1::GFP at the synaptic tiling border. At least, we observe no clear reduction in the overall expression level of PLX-1::GFP in *unc-4* mutants.

**Author response image 2. respfig2:** 

To test if *unc-4* and *unc-37* controls the expression of *plx-1*, we examined the expression of P*plx-1::gfp* in *unc-4* and *unc-37* mutants, and found that there was slight reduction in % animals expressing P*plx-1::gfp* in the anterior DA8/9 of *unc-37* mutants (Figure 1—figure supplement 4). Given the fully penetrant synaptic tiling defects, we do not think that slight reduction in *plx-1* expression is responsible for the synaptic tiling defects of *unc-4* and *unc-37* mutants. We also found that the expression of *plx-1* cDNA under the DA neuron-specific promoter (P*unc-4c*) did not rescue the synaptic tiling defects of *unc-4* and *unc-37* mutants (not shown). Therefore, the loss or reduction of *plx-1* expression is unlikely the mechanism underlying the synaptic tiling defects of *unc-4* and *unc-37.*

4) In Figure 2+3, the authors drew the conclusion that unc-4 is not required during embryogenesis for synaptic connection. Since they imaged the animals during L4, it is possible that the unc-4 is required for both early and late for synaptic tiling. They should score the phenotype in late L1 or L2 to make sure that the phenotype is not there in the embryonic KD experiments.

We examined the synaptic tiling defects of the embryonic knockdown of *unc-4* and *unc-37* at early L2 stage. Consistent with our observation at L4 stage animals, we found that embryonic knockdown of *unc-37* but not *unc-4* caused synaptic tiling defects at L2 stage. This result further strengthens our conclusion that *unc-4* is not required during DA neurogenesis at embryonic stage to control their synaptic tiling. The data is included in Figure 4.

Reviewer #3 (Recommendations for the authors):Do the RAB-3::GFPnovo and RAB-3::mCherry markers always have complete overlap, for example in DA9? I'm wondering if variability in the extent of labeling across the 2 markers could influence the ability to assess tiling defects, for example in cases where the RAB-3::GFP marker is only partially degraded.

We have answered this comment in the essential revision #2

The previously defined roles for unc-4/unc-37 focused around AVA connectivity with post-embryonic born VA neurons. The authors provide evidence that a subset of neuronal identity markers in DA8 and DA9 neurons are largely unchanged in unc-4 and unc-37 mutants. Do the authors envision that unc-4 and unc-37 have distinct roles in embryonic born DA neurons from those in post embryonic born VA neurons? Or perhaps additional roles?

The lack of major cell fate defects in the DA8 and DA9 neurons of *unc-4* and *unc-37* mutants was surprising to us, because previous work indeed observed that the cell fate defects in the DA neurons in the ventral nerve cord (DA2-DA7) in *unc-4* mutants (Winnier et al., 1999; Kerk et al., 2017). One possible explanation is that the mechanisms of cell fate specification of these two posterior DA neurons are slightly different from the rest of DA and VA neurons. Consistent with this idea, DA8 and DA9 exhibit distinct gene expression profile compared with anterior DA neurons (Kratsios et al., 2017). In the revised manuscript, we described this possibility in the discussion.

The downstream pathways (ceh-12, wnt) identified as important for tiling strongly overlap with those previously identified as important for neuronal fate determination. While perhaps not absolutely critical, identification of the proposed additional Wnt-regulated TF that is important (with ceh-12) for tiling might distinguish specific mechanisms for regulation of tiling and be a strong addition the story.

We agree that identifying the additional transcription factor(s) that function downstream of Wnt signaling would greatly enhance our understanding of transcriptional regulation of precise synapse pattern formation. However, given the genetic redundancy between *ceh-12* and these factor(s), identification of these genes would be challenging. We hope to identify them through single cell RNA sequencing of DA8/9 in *unc-4* mutants, as well as genetic suppressor screening of the synaptic tiling defects in the *unc-4; ceh-12* mutants in the future. We included these future experiments in the revised discussion.

Related, has tiling been examined in VA neurons? Might unc-4/unc-37 have roles in both neuronal fate determination and later roles in synaptic tiling in these neurons? I don't expect the authors to resolve these questions experimentally, but a more direct discussion might help to clarify.

Our synaptic tiling marker is supposed to label two posterior VA neurons, VA11 and VA12, with GFPnovo2::RAB-3 and mCherry::RAB-3. However, the synaptic tiling between VA11 and VA12 labeled with *mizIs3* (and *wyIs446*) are highly variable in wildtype, which made the phenotypic examination of *unc-4* and *unc-37* difficult. We hope to be able to answer this important question with a better VA-synaptic tiling marker in the future. In the revised manuscript, we have included a discussion regarding the potential functions of *unc-4* and *unc-37* in VA synaptic tiling and AVA-VA synaptic specificity.